# COMMUNICATION-EFFICIENT FEDERATED LEARNING WITH ACCELERATED CLIENT GRADIENT

## ABSTRACT

Federated learning often suffers from slow and unstable convergence due to heterogeneous characteristics of participating client datasets. Such a tendency is aggravated when the client participation ratio is low since the information collected from the clients is prone to have large variations. To tackle this challenge, we propose a novel federated learning framework, which improves the consistency across clients and facilitates the convergence of the server model. This is achieved by making the server broadcast a global model with a gradient acceleration. By adopting the strategy, the proposed algorithm conveys the projective global update information to participants effectively with no extra communication cost, and relieves the clients from storing the previous models. We also regularize local updates by aligning each of the client with the overshot global model to reduce bias and improve the stability of our algorithm. We perform comprehensive empirical studies on real data under various settings and demonstrate remarkable performance gains of the proposed method in terms of accuracy and communication efficiency compared to the state-of-the-art methods, especially with low client participation rates. We will release our code to facilitate and disseminate our work.

## 1 INTRODUCTION

Federated learning (McMahan et al., 2017) is a large-scale machine learning framework that learns a shared model in a central server through collaboration with a large number of remote clients with separate datasets. This decentralized learning concept allows federated learning to achieve the basic level of data privacy since the server does not observe training data directly. On the other hand, remote clients such as mobile or IoT devices have limited communication bandwidths, and federated learning algorithms are particularly sensitive to communication costs.

A baseline algorithm of federated learning, FedAvg (McMahan et al., 2017) updates a subset of its client models based on a gradient descent method using their local data and then uploads the resulting models to the server for computing the global model parameters via model averaging. As discussed extensively on the convergence of FedAvg (Stich, 2019; Yu et al., 2019; Wang & Joshi, 2021; Stich & Karimireddy, 2019; Basu et al., 2020), multiple local updates conducted before server-side aggregation provide theoretical support and practical benefit of federated learning by reducing communication cost greatly.

Despite the initial success, federated learning faces two key challenges: high heterogeneity in training data distributed over clients and limited participation rates of clients. Several studies (Zhao et al., 2018; Karimireddy et al., 2020) have shown that multiple local updates in the clients with non-*i.i.d* (independent and identically distributed) data lead to client model drift, in other words, diverging updates in the individual clients. Such a phenomenon introduces the high variance issue in the FedAvg step for global model updates, which hampers the convergence to the optimal average loss over all clients (Li et al., 2020; Wang et al., 2019b; Khaled et al., 2019; Li et al., 2019b; Hsieh et al., 2020; Wang et al., 2020). The challenge related to client model drift is exacerbated when the client participation rate per communication round is low, due to unstable client device operations and limited communication channels.

To properly address the client heterogeneity issue, we propose a novel optimization algorithm for federated learning, Federated averaging with Accelerated Client Gradient (FedACG), which conveys

the momentum of the global gradient to clients and enables the momentum to be incorporated into the local updates in the individual clients. Specifically, we introduce an extra-gradient step on the global model via the global momentum, which allows each client performs its local gradient step along the future gradient. This approach turns out to be effective for reducing the gap between global and local losses. Contrary to the existing methods that require to send additional bits to communicate the momentum, FedACG transmits the global model integrated with the momentum in the form of a single message and saves the cost for communication. In addition, FedACG adds a regularization term in the objective function of clients to make the local gradients more consistent across clients.

Although there have been a growing number of works that handle the client heterogeneity in federated learning, FedACG has the following major advantages. Unlike existing approaches focusing on server-level optimization (Reddi et al., 2021; Wang et al., 2019a; Hsu et al., 2019) or client-level optimization (Xu et al., 2021; Acar et al., 2021; Karimireddy et al., 2020; Li et al., 2021; 2020; Zhang et al., 2020; Karimireddy et al., 2021; Li et al., 2019a; Liang et al., 2019), FedACG incorporates the momentum based on the global gradient information for client-side updates. This strategy allows the proposed algorithm to achieve the same level of task-specific performance with fewer communication rounds. Moreover, while most of existing methods have additional requirements compared to FedAvg including full participation (Liang et al., 2019; Zhang et al., 2020; Khanduri et al., 2021), additional communication bandwidth (Xu et al., 2021; Karimireddy et al., 2020; Zhu et al., 2021; Karimireddy et al., 2021; Li et al., 2019a; Das et al., 2020; Gao et al., 2022), and memory budgets in clients to store local states or variables (Acar et al., 2021; Karimireddy et al., 2020; Li et al., 2021; Gao et al., 2022), FedACG is completely free from any additional communication and memory overhead, which ensures the compatibility with large-scale and low-participation federated learning problems. The main contributions of this paper are summarized as follows.

- We propose a communication-efficient federated optimization algorithm that deals with client heterogeneity effectively. The proposed approach employs the global momentum for the acceleration of client gradients to facilitate the optimization of local models.

- We also revise the objective function of clients, which augments a regularization term to the local gradient direction, which further aligns the gradients of server and individual clients.

- We show that the proposed approach does not require any additional communication cost and memory overhead, which is desirable for the real-world settings of federated learning.

- We demonstrate outstanding performance of our optimization technique in terms of communication efficiency and robustness to client heterogeneity, especially when the participation ratio is low.

## 2 RELATED WORK

Federated learning was first introduced in McMahan et al. (2017), which formulates the problem and provides the FedAvg algorithm as a solution for its key challenges such as non-iid client data, massively distributed clients, and partial participation of clients. Many works explore the negative influence of heterogeneity in federated learning empirically (Zhao et al., 2018) and derive convergence rates depending on the level of heterogeneity (Li et al., 2020; Wang et al., 2019b; Khaled et al., 2019; Li et al., 2019b; Hsieh et al., 2020; Wang et al., 2020).

There exists a long line of research for client-side optimization to prevent the divergence of clients from the global model. FedProx (Li et al., 2020) penalizes the difference between the server and client parameters, while FedDyn (Acar et al., 2021) and FedPD (Zhang et al., 2020) use cumulative gradients of each client to dynamically regularize local update. FedDC (Gao et al., 2022) introduces the auxiliary drift variables for each client to reduce the impact of the local drift on the global objective. There is another line of works which adopt variance reduction techniques in client update to eliminate inconsistent update across clients. SCAFFOLD (Karimireddy et al., 2020) and Mime (Karimireddy et al., 2021) employ control variates for local updates while FedDANE (Li et al., 2019a) and FedCM (Xu et al., 2021) add a gradient correction term based on the server gradient. FedPA (Al-Shedivat et al., 2021) de-bias client updates by estimating the global posterior on the client side. On the other hand, some approaches adopt a contrastive loss (Li et al., 2021), knowledge distillation (Kim et al., 2022), or a generative model (Zhu et al., 2021) to ensure the similarity of the representations between the global model and local networks. FedSAM (Qu et al.,

---

**Algorithm 1** FedACG

---

**Input:** $\beta$, $\lambda$, initial server model $\theta^0$, number of clients $N$, number of communication rounds $T$,
       local learning rate $\eta$
Initialize the global momentum, $m^0 = 0$.
**for** each round $t = 1, 2, \ldots, T$ **do**
    Sample a subset of clients $S_t \subseteq \{1, \ldots, N\}$.
    Server sends $\theta^{t-1} + \lambda m^{t-1}$ to initialize local models for all clients $i \in S_t$.
    **for** each client $i \in S_t$, **in parallel do**
        Set $\theta_i^t = \underset{\theta}{\operatorname{argmin}} \, \mathcal{L}_i(\theta) + \frac{\beta}{2}\|\theta - (\theta^{t-1} + \lambda m^{t-1})\|^2$
        Client sends $\theta_i^t$ back to the server.
    **end**
    **In server:**
        $\theta^t = \sum_{i \in S_t} \omega_i \theta_i^t$
        $m^t = \theta^t - \theta^{t-1}$
**end**
**Return** $\theta^t$

---

2022) and ASAM (Caldarola et al., 2022) apply SAM (Foret et al., 2021) as a client-side optimizer for reducing the gap between global and local losses. However, most of these methods require full participation (Zhang et al., 2020; Khanduri et al., 2021), additional communication cost (Xu et al., 2021; Karimireddy et al., 2020; Zhu et al., 2021; Karimireddy et al., 2021; Li et al., 2019a; Das et al., 2020; Gao et al., 2022), or extra client storage (Acar et al., 2021; Karimireddy et al., 2020; Li et al., 2021; Gao et al., 2022), which can be problematic in realistic federated learning tasks.

Server-side optimization techniques also have been explored for the stability and speedup of convergence. These approaches adopt a momentum SGD (Hsu et al., 2019), or an adaptive gradient-descent method (Reddi et al., 2021; Caldarola et al., 2022), while FedDF (Lin et al., 2020) utilizes the averaged representations of local models on proxy data for aggregation. STEM (Khanduri et al., 2021) and FedGLOMO (Das et al., 2020) apply STORM algorithm (Cutkosky & Orabona, 2019) to both server-level and client-level SGD procedures for reducing high variance in server model update.

Meanwhile, another set of works aims to decrease the communication cost per round by compressing the model transmitted. FedPAQ (Reisizadeh et al., 2020), FedCAMS (Wang et al., 2022) and FedCOMGATE (Haddadpour et al., 2021) quantize the communicated message by using low bit precision, while FedPara (Nam et al., 2022) use low-rank Hadamard product to reparameterize the model's weights. These works are orthogonal to our approach, so they can be readily combined with our proposed method.

## 3 METHOD

In the federated learning setting, there are $N$ clients that optimize their local models based on the corresponding private datasets as well as a central server that broadcasts the global model and then aggregates messages from the clients. Let $\mathcal{L}_i(\theta) := \mathbb{E}_{(x,y) \sim \mathcal{D}_i}[\ell_i((x,y);\theta)]$ be the loss function of the $i^{\text{th}} \in \{1, \ldots, N\}$ client with a local dataset denoted by $\mathcal{D}_i$. Then, our goal is to train a model that minimizes the average loss of all clients as follows:

$$\min_{\theta} \left\{ \mathcal{L}(\theta) := \sum_{i=1}^{N} \omega_i \mathcal{L}_i(\theta) \right\}, \tag{1}$$

where $\theta$ is the parameter of the global model and $\omega_i$ is the normalized weight of the $i^{\text{th}}$ client proportional to the size of the local dataset. We focus on the non-*i.i.d* data setting, where local datasets have heterogeneous distributions. Note that the communication of training data between clients and the central server is strictly prohibited in principle due to privacy.

### 3.1 FEDACG

To reduce the inconsistency between the local models and the consequent divergence of the global one, we incorporate the global momentum into local models for guiding local updates.

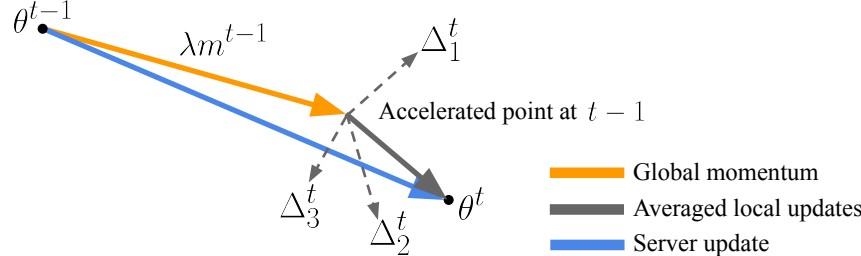

Figure 1: An illustration of the proposed accelerated client gradient method. We first partially update the global model in the direction of the global momentum (orange) and then aggregate local updates (gray), resulting in the server model in the next round (blue). Through this anticipatory update, we make the individual local updates aligned with the global gradient and achieve speed-up of convergence.

**Overall framework** Each round of FedAGC starts from the server. In the $t^{\text{th}}$ communication round, the server computes its momentum $m^{t-1} := \theta^{t-1} - \theta^{t-2}$, and broadcasts the accelerated global model $\theta^{t-1} + \lambda m^{t-1}$ as a single message to the active client set $\mathcal{S}_t \subseteq \{1, \dots, N\}$, where $\lambda \in (0, 1]$ controls the importance of the global momentum. Each participating client optimizes its local model from the momentum-integrated initialization. The objective of each client is to minimize the sum of its empirical loss on the local data and a penalty from the difference between the local online model and the accelerated global model, which is given by

$$\min_{\theta_i^t} \left\{ \mathcal{F}_i(\theta_i^t) := \mathcal{L}_i(\theta_i^t) + \frac{\beta}{2} \|\theta_i^t - (\theta^{t-1} + \lambda m^{t-1})\|^2 \right\}, \tag{2}$$

where $\beta$ controls the balance between the two terms. Each client uploads their trained model $\theta_i^t$ to the server, and then the server constructs the next server model $\theta^t$ via a simple aggregation, *i.e.*, $\theta^t = \sum_{i \in \mathcal{S}_t} \omega_i \theta_i^t$. Algorithm 1 presents the procedure of FedACG.

**Accelerated client gradient** The main idea of this work, accelerated client gradient, is to leverage the global momentum and allow clients to look ahead the landscape of the global loss. The momentum $m^t$ serves as an approximate gradient of the global loss since it maintains past global updates even in partial participating setting. Let the local update of each participating client be $\Delta_i^t = \theta_i^t - (\theta^{t-1} + \lambda m^{t-1})$. Then, $m^t$ is defined recursively with an exponential decay factor $\lambda$ as

$$m^t = \theta^t - \theta^{t-1} = \left[ \Delta^t + (\theta^{t-1} + \lambda m^{t-1}) \right] - \theta^{t-1} = \Delta^t + \lambda m^{t-1}, \tag{3}$$

where $\Delta^t = \sum_{i \in S_t} \omega_i \Delta_i^t$ denotes the expected local updates of all participating clients in the current round $t$.

As illustrated in Figure 1, FedACG makes an anticipatory update by integrating $\lambda m^{t-1}$ to the previous global model $\theta^{t-1}$. This strategy allows the updates of each client to be aligned with the trajectory of the global gradients, which improves consistency of local updates in FedACG. Our approach has a similar motivation with meta-learning (Finn et al., 2017), where a meta-learner identifies the optimal point to facilitate the optimization of all target tasks.

**Regularization with momentum-integrated model** In addition to the initial acceleration for local training, the second term of our local objective function in Eq. (2) takes the advantage of the global gradient information to reduce the variations of client-specific gradients, $\Delta_i^t$. This regularization term enforces the local model not to deviate from the accelerated point, preventing each client from falling into biased local minima.

### 3.2 DISCUSSION

While our formulation has something in common with the existing works that also address client heterogeneity by employing global gradient information for the local update, FedACG has the following major advantages. First, contrary to Karimireddy et al. (2020); Xu et al. (2021); Gao et al. (2022), the server and clients only communicate model parameters without imposing additional network

overhead for transmitting gradients and other information; the server broadcasts $(\theta^{t-1} + \lambda m^{t-1})$ as a single message and each client sends $\theta_{i,K}^t$ to the server. This is a critical benefit because the increase in communication cost challenges many realistic federated learning applications involving clients with limited network bandwidths. Second, FedACG is robust to the low participation rate of clients and allows new-arriving clients to join the training process immediately without a warmup phase because, unlike Karimireddy et al. (2020), Acar et al. (2021), Li et al. (2021), and Gao et al. (2022), the clients are supposed to neither store their local states nor use them for model updates.

### 3.3 Convergence analysis of FedACG

We now present the theoretical convergence rate of FedACG. We first state two assumptions for the local loss functions $\ell_i(\theta)$, which are commonly used in several previous works on federated optimization (Karimireddy et al., 2020; Reddi et al., 2021; Xu et al., 2021; Acar et al., 2021). First, the local function $\ell_i(\cdot)$ is assumed to be $L$-smooth for all $i \in \{1, \dots, N\}$, *i.e.*,

$$\|\nabla \ell_i(x) - \nabla \ell_i(y)\| \le L\|x - y\| \quad \forall x, y. \tag{4}$$

Second, if we additionally assume the convexity of the functions $\{\ell_i(\cdot)\}_{i=1}^N$, we have

$$\forall x, \qquad \frac{1}{2LN} \sum_{i=1}^N \|\nabla \ell_i(x) - \nabla \ell_i(x_*)\|^2 \le \ell(x) - \ell(x_*) \text{ and} \tag{5}$$

$$\forall x, y, z, \ \langle \nabla \ell_i(x), z - y \rangle \le -\ell_i(z) + \ell_i(y) + \frac{L}{2}\|z - x\|^2, \tag{6}$$

where $\ell(x) = \frac{1}{N} \sum_{i=1}^N \ell_i(x)$ and $\nabla \ell(x_*) = 0$. Based on the above assumptions, we derive the following asymptotic convergence bound of FedACG. Note that we make no further assumptions such as a form of bounded variance and gradients used in Karimireddy et al. (2020), Reddi et al. (2021), and Xu et al. (2021).

**Theorem 1.** *Assuming the convexity and L-smoothness of $\{\ell_i(\cdot)\}_{i=1}^N$, for $\frac{1}{2} < \lambda < 1$, Algorithm 1 satisfies*

$$\mathbb{E}\left[\ell\left(\frac{1}{T}\sum_{t=1}^T \theta^{t-1}\right) - \ell(\theta_*)\right] \le \frac{\sqrt{\lambda(1-\lambda)}}{T}\left(L\|\theta^0 - \theta_*\|^2 + \frac{1}{LN}\sum_{i=1}^N \|\nabla \ell_i(\theta_i^0)\|^2\right),$$

*where $\theta_* = \underset{\theta}{\mathrm{argmin}}\ \ell(\theta)$ and $\theta^t = \frac{1}{|S_t|}\sum_{i \in S_t} \theta_i^t$.*

Theorem 1 implies that, for convex and smooth local functions, the global objective function is expected to converge at a rate of $O\left(\frac{\sqrt{\lambda(1-\lambda)}}{T}\right)$. This rate is the empirical loss averaged over all devices. It further implies a higher value of $\lambda$ improves the convergence rate under the convex setting, which will be empirically verified in our experiments. Please, refer to the supplementary document for the full proof.

## 4 Experiments

This section presents empirical evaluations of FedACG and competing federated learning methods, to highlight the robustness to data heterogeneity of the proposed method in terms of performance and communication-efficiency.

### 4.1 Experimental setup

**Datasets and baselines** We conduct a set of experiments on CIFAR-10 (Krizhevsky et al., 2009), CIFAR-100 (Krizhevsky et al., 2009), and Tiny-ImageNet[1] (Le & Yang, 2015) with various data heterogeneity levels and participation rates. Note that Tiny-ImageNet (200 classes with $10,000$ samples) is more natural and realistic compared to the simple datasets, such as MNIST and CIFAR,

---

[1] https://www.kaggle.com/c/tiny-imagenet

Table 1: Comparisons of FedACG with baselines on CIFAR-10, CIFAR-100 and Tiny-ImageNet for two different federated learning settings. For a moderate-scale experiment (a), the number of clients and participation rate, are set to 100, and 5%, respectively, while a large-scale experiment (b) has 500 clients with 2% participation rate. The Dirichlet parameter is commonly set to 0.3. Accuracy at target round and the communication round to reach target test accuracy is based on running exponential moving average with parameter 0.9. The arrows indicate whether higher (↑) or lower (↓) is better. The best performance in each column is denoted in **bold**. FedCM[†] and FedDC[‡] require $1.5\times$ and $2\times$ communication cost for each communication round, respectively.

(a) Moderate-scale: 5% participation, 100 clients

| Method | CIFAR-10 | | | | CIFAR-100 | | | | Tiny-ImageNet | | | |
|---|---|---|---|---|---|---|---|---|---|---|---|---|
| | Acc. (%, ↑) | | Rounds (↓) | | Acc. (%, ↑) | | Rounds (↓) | | Acc. (%, ↑) | | Rounds (↓) | |
| | 500R | 1000R | 81% | 85% | 500R | 1000R | 47% | 55% | 500R | 1000R | 35% | 38% |
| FedAvg (McMahan et al., 2017) | 74.36 | 82.53 | 840 | 1000+ | 41.88 | 47.83 | 924 | 1000+ | 33.94 | 35.37 | 645 | 1000+ |
| FedProx (Li et al., 2020) | 73.70 | 82.68 | 826 | 1000+ | 42.43 | 48.32 | 881 | 1000+ | 34.14 | 35.53 | 613 | 1000+ |
| FedAvgM (Hsu et al., 2019) | 80.56 | 85.48 | 519 | 828 | 46.98 | 53.29 | 515 | 1000+ | 36.32 | 38.51 | 416 | 829 |
| FedADAM (Reddi et al., 2021) | 72.33 | 81.73 | 908 | 1000+ | 44.80 | 52.48 | 691 | 1000+ | 33.22 | 38.91 | 658 | 945 |
| FedDyn (Acar et al., 2021) | 84.82 | 88.10 | 392 | 646 | 48.38 | 55.79 | 424 | 883 | 37.35 | 41.18 | 344 | 573 |
| MOON (Li et al., 2021) | 83.32 | 86.30 | 371 | 686 | 53.15 | 58.37 | 284 | 640 | 36.62 | 40.33 | 410 | 627 |
| FedCM[†] (Xu et al., 2021) | 78.92 | 83.71 | 624 | 1000+ | 52.44 | 58.06 | 293 | 747 | 31.61 | 37.87 | 694 | 1000+ |
| FedDC[‡] (Gao et al., 2022) | **86.52** | 87.47 | 323 | 519 | 54.25 | 59.01 | 333 | 553 | 40.32 | 45.51 | 340 | 403 |
| FedACG (ours) | 85.13 | **89.10** | **319** | **450** | **55.79** | **62.51** | **260** | **409** | **42.26** | **46.31** | **226** | **331** |

(b) Large-scale: 2% participation, 500 clients

| Method | CIFAR-10 | | | | CIFAR-100 | | | | Tiny-ImageNet | | | |
|---|---|---|---|---|---|---|---|---|---|---|---|---|
| | Acc. (%, ↑) | | Rounds (↓) | | Acc. (%, ↑) | | Rounds (↓) | | Acc. (%, ↑) | | Rounds (↓) | |
| | 500R | 1000R | 73% | 77% | 500R | 1000R | 36% | 40% | 500R | 1000R | 24% | 30% |
| FedAvg (McMahan et al., 2017) | 58.74 | 71.45 | 1000+ | 1000+ | 30.16 | 38.11 | 842 | 1000+ | 23.63 | 29.48 | 523 | 1000+ |
| FedProx (Li et al., 2020) | 57.88 | 70.75 | 1000+ | 1000+ | 29.28 | 36.16 | 966 | 1000+ | 25.45 | 31.71 | 445 | 799 |
| FedAvgM (Hsu et al., 2019) | 65.85 | 77.49 | 753 | 959 | 31.80 | 40.54 | 724 | 955 | 26.75 | 33.26 | 386 | 687 |
| FedADAM (Reddi et al., 2021) | 61.53 | 69.94 | 1000+ | 1000+ | 24.40 | 30.83 | 1000+ | 1000+ | 21.88 | 28.08 | 648 | 1000+ |
| FedDyn (Acar et al., 2021) | 65.49 | 77.92 | 732 | 936 | 31.58 | 41.01 | 691 | 927 | 24.35 | 29.54 | 483 | 1000+ |
| MOON (Li et al., 2021) | 69.15 | 78.06 | 617 | 872 | 33.51 | 42.41 | 601 | 828 | 26.69 | 31.81 | 382 | 741 |
| FedCM[†] (Xu et al., 2021) | 69.27 | 76.57 | 742 | 1000+ | 27.23 | 38.79 | 872 | 1000+ | 19.41 | 24.09 | 975 | 1000+ |
| FedDC[‡] (Gao et al., 2022) | 71.86 | **83.49** | 518 | 686 | 34.64 | 45.93 | 569 | 741 | 25.72 | 28.92 | 420 | 1000+ |
| FedACG (ours) | **73.61** | 82.80 | **484** | **605** | **35.68** | **48.40** | **505** | **616** | **31.47** | **38.48** | **246** | **447** |

used for evaluation of many previous methods (McMahan et al., 2017; Karimireddy et al., 2020). We generate IID data split by randomly assigning training data to individual clients without replacement. For the non-IID data, we simulate the data heterogeneity by sampling the label ratios from a Dirichlet distribution with a symmetric parameter $\{0.3, 0.6\}$, following Hsu et al. (2019). We keep the training data balanced, so each client holds the same amount of data.

We compare our method, FedACG, with several state-of-the-art federated learning techniques, which include FedAvg (McMahan et al., 2017), FedProx (Li et al., 2020), FedAvgM (Hsu et al., 2019), FedADAM (Reddi et al., 2021), FedDyn (Acar et al., 2021), FedCM (Xu et al., 2021), MOON (Li et al., 2021), FedDC (Gao et al., 2022). We adopt a standard ResNet-18 (He et al., 2016) as backbone network for all benchmarks, but we replace batch normalization by group normalization as suggested in Hsieh et al. (2020).

**Evaluation metrics** To evaluate the generalization performance of the methods on the global distribution, we use the entire test set in the CIFAR-10, CIFAR-100, and Tiny-ImageNet. Since both the speed of learning as well as the final performance are important quantities for federated learning, we measure: (i) the performance attained at a specified number of rounds, and (ii) the number of rounds needed for an algorithm to attain the desired level of target accuracy, following Al-Shedivat et al. (2021). For the selection of target accuracies, we first choose the median of all methods at round 1000 and another representative value lower than the median. For methods that could not achieve aimed accuracy within the maximum communication round, we append the communication round with a + sign.

Table 2: Effect of low participation rate, 1% over 500 clients with Dirichlet (0.3) split, for FedACG and the baselines on CIFAR-10 and CIFAR-100. Accuracy at the target round and the communication round to reach target test accuracy are based on running exponential moving average with parameter 0.9. The arrows indicate whether higher ($\uparrow$) or lower ($\downarrow$) is better. FedCM[†] and FedDC[‡] require $1.5\times$ and $2\times$ communication cost for each communication round, respectively.

| Method | CIFAR-10 | | | | CIFAR-100 | | | |
|---|---|---|---|---|---|---|---|---|
| | accuracy (%, $\uparrow$) | | rounds (#, $\downarrow$) | | accuracy (%, $\uparrow$) | | rounds (#, $\downarrow$) | |
| | 500R | 1000R | 64% | 68% | 500R | 1000R | 30% | 35% |
| FedAvg (McMahan et al., 2017) | 54.71 | 68.96 | 792 | 949 | 26.94 | 35.69 | 636 | 950 |
| FedProx (Li et al., 2020) | 55.18 | 69.80 | 773 | 919 | 26.92 | 35.41 | 648 | 963 |
| FedAvgM (Hsu et al., 2019) | 57.82 | 71.12 | 669 | 812 | 29.29 | 39.36 | 530 | 755 |
| FedADAM (Reddi et al., 2021) | 47.97 | 55.11 | 1000+ | 1000+ | 17.72 | 23.92 | 1000+ | 1000+ |
| FedDyn (Acar et al., 2021) | 54.86 | 70.78 | 713 | 858 | 27.86 | 36.31 | 595 | 896 |
| MOON (Li et al., 2021) | **64.55** | 73.89 | 491 | 645 | 28.29 | 36.37 | 567 | 886 |
| FedCM[†] (Xu et al., 2021) | 49.21 | 60.38 | 1000+ | 1000+ | 16.32 | 22.59 | 1000+ | 1000+ |
| FedDC[‡] (Gao et al., 2022) | 60.56 | 75.06 | 610 | 681 | 29.14 | 38.84 | 519 | 789 |
| FedACG (ours) | 63.70 | **76.45** | 497 | **618** | **31.74** | **45.18** | **458** | **581** |

**Implementation details** We use PyTorch (Paszke et al., 2019) to implement FedACG and the other baselines. We follow Acar et al. (2021); Xu et al. (2021) for evaluation protocol. For local update, we use the SGD optimizer with a learning rate 0.1 for all approaches on the three benchmarks. We apply exponential decay on the local learning rate, and the decay parameter is selected from $\{1.0, 0.998, 0.995\}$. We apply no momentum for local SGD, but apply weight decay of 0.001 to prevent overfitting. We also use gradient clipping to increase the stability of the algorithms. The number of local training epochs over each client update is set to 5, and the batch size is set so that the total iteration for local updates is set to 50 for all experiments. We set the global learning rate as 1 for all methods except for FedADAM which is set to 0.01. We list the details of the hyperparameters specific to FedACG and the compared algorithms in Appendix B.

## 4.2 MAIN RESULTS

**Evaluation with standard federated learning scenarios** We first present the performance of the proposed approach, FedACG, on CIFAR-10, CIFAR-100, and Tiny-ImageNet in the scenarios by varying the number of clients, data heterogeneity, and participation rate. Our experiment has been performed on two different settings; one is with a moderate-scale, which involves 100 devices with 5% participation rate per round, and the other is with a large number of clients, 500 with 2% participation rate. Note that the number of clients in the large-scale setting is 5 times more than the moderate-scale experiment, which reduces the number of examples per client by 80%.

Table 1a demonstrates that FedACG improves accuracy and convergence speed significantly and consistently compared with other federated learning methods in most cases. This is partly because FedACG enables each client to look ahead the global update and aligns the local model updates with the global gradient trajectory. Note that FedCM and FedDC require $1.5\times$ and $2\times$ communication costs for each communication round respectively since they communicate the current model and the associated gradient information per round, while others only require model parameters.

For the large-scale setting, Table 1b illustrates the outstanding performance of FedACG on CIFAR-10, CIFAR-100, and Tiny-ImageNet, except for the accuracy at 1K rounds on CIFAR-10. One noticeable thing is that the overall performance is lower than the case with a moderate number of clients. This is because the number of training data for each client decreases and each client suffers more from the heterogeneous data distribution. Nevertheless, we observe that FedACG outperforms other methods consistently in most cases; the accuracy gap between FedACG and its strongest competitor becomes larger in these more challenging scenarios. The results from the large-scale experiments informs the robustness of FedACG to the heterogeneity and limited participation of clients. We present more comprehensive results for the convergence of FedACG in Appendix D.1.

**Effect of low participation rate** Partial participation is a critical challenge to slow down the convergence of the global model in federated learning. To verify the robustness to the low participation rate of clients, we perform experiments when the total number of clients is 500 and the participation

Table 3: Results on CIFAR-100 when client set changes dynamically: we sample 250 clients out of 500 clients as a candidate clients set at every 100 rounds over 10 stages on Dirichlet (0.3) split. 10 clients out of the sampled client set participate for the local training for each communication round. FedDC† requires 2× communication cost for each communication round.

| Method | CIFAR-100 | | | |
| | Acc. (%, ↑) | | Rounds (↓) | |
| | 500R | 1000R | 30% | 38% |
|---|---|---|---|---|
| FedAvg (McMahan et al., 2017) | 28.61 | 35.87 | 577 | 1000+ |
| FedDyn (Acar et al., 2021) | 29.45 | 38.47 | 517 | 941 |
| MOON (Li et al., 2021) | 30.88 | 39.57 | 430 | 852 |
| FedDC† (Gao et al., 2022) | 31.35 | 36.82 | 469 | 1000+ |
| FedACG (ours) | **32.70** | **41.51** | **376** | **769** |

Table 4: Contribution of individual components in FedACG at $1000^{th}$ rounds on CIFAR-10 and CIFAR-100 with 2% participation and 500 clients.

| Accelerated gradient ($\lambda$) | Regularization term ($\beta$) | CIFAR-10 | CIFAR-100 |
|---|---|---|---|
| | | 71.45 | 38.11 |
| | ✓ | 70.75 | 36.16 |
| ✓ | | 82.20 | 46.80 |
| ✓ | ✓ | **82.80** | **48.40** |

rate is as low as 1%. The numbers of local epochs and iterations are set to 5 and 50, respectively. This setting makes each client have very few training examples and increases client heterogeneity significantly. Table 2 again shows that FedACG has the best performance for most cases. Note that the performance gap between FedACG and the second-best method, FedDC, becomes larger than when the participation rate is 2%: from -0.69%p to 1.39%p on CIFAR-10 and from 2.47%p to 6.34%p on CIFAR-100 at round 1000. This is partly because the local states managed by FedDC are susceptible to get stale quickly in this scenario, making its convergence require extra iterations. In contrast, our method does not rely on the past information stored in local devices and is not affected by this issue.

**Evaluation on dynamic client set**  Since FedACG is free from the requirements of storing local model history for local updates, it is conceptually better-suited for the scenarios in the presence of newly participating clients. In order to validate the property, we conduct an experiment on CIFAR-100 with 500 clients for Dirichlet(0.3) splits. We sample 250 clients at every 100 rounds as a candidate client set, and then 10 randomly sampled clients (4% of clients) participate in the local training for each communication round. Table 3 shows that FedACG outperforms FedAvg and FedDyn. Note that FedDyn is worse than FedAvg since the client model has trouble with its heterogeneity and divergence because new clients have no or non-informative local states.

### 4.3 ABLATION STUDY

**Contribution of individual components**  Table 4 presents the contribution of individual components in the experiment on CIFAR-10 for the large-scale federated learning setting. We observe that the accelerated client gradient for local training has more critical impact on accuracy with 1000 rounds. Note that the proposed regularization term in local loss function shows larger performance gain when used with the accelerated client gradient, while employing the regularization term only do not necessarily achieve performance gains in CIFAR-10 and CIFAR-100.

**Ablation study for hyperparameters**  Table 5 presents the accuracy of FedACG for Dirichlet(0.3) and IID splits by varying the value of $\lambda$ and $\beta$, which control the momentum integration of the server model and the weight of the proximal term, respectively. As shown in the table 5a, the low values of $\lambda$ do not work well, supporting the benefit of the proposed accelerated client gradient strategy, while Table 5b shows that the accuracy is stable with respect to $\beta$.

Table 5: Ablation study of FedACG to the weights of the two hyperparameters, $\lambda$ (a) and $\beta$ (b), with respect to the accuracy at $1000^{\text{th}}$ rounds on CIFAR-10 in 2% participation and 500 clients.

(a) Sensitivity of FedACG to $\lambda$

| $\lambda$ | 0 | 0.25 | 0.5 | 0.75 | 0.85 | 0.95 |
|---|---|---|---|---|---|---|
| Dir(0.3) | 70.75 | 73.93 | 79.14 | 81.32 | 82.80 | 78.25 |
| IID | 72.20 | 74.29 | 80.52 | 85.52 | 86.83 | 84.37 |

(b) Sensitivity of FedACG to $\beta$

| $\beta$ | 0.001 | 0.01 | 0.1 | 1 |
|---|---|---|---|---|
| Dir(0.3) | 82.10 | 82.80 | 82.32 | 82.44 |
| IID | 86.54 | 86.83 | 86.72 | 85.92 |

Table 6: Results on the realistic federated learning datasets which contain feature skewness and data imbalance between the clients. FedCM[†] and FedDC[‡] require $1.5\times$ and $2\times$ communication cost for each communication round, respectively.

| Method | FEMNIST | | CelebA | |
|---|---|---|---|---|
| | Acc. (%, ↑) | Rounds (↓) | Acc. (%, ↑) | Rounds (↓) |
| | 500R | 78% | 500R | 88% |
| FedAvg (McMahan et al., 2017) | 78.38 | 328 | 89.92 | 134 |
| FedProx (Li et al., 2020) | 78.34 | 328 | 89.90 | 132 |
| FedAvgM (Hsu et al., 2019) | 78.37 | 256 | 89.85 | 113 |
| FedADAM (Reddi et al., 2021) | 75.96 | 500+ | 87.00 | 500+ |
| FedDyn (Acar et al., 2021) | 79.80 | 227 | 89.74 | 126 |
| MOON (Li et al., 2021) | 78.33 | 336 | 87.95 | 500+ |
| FedCM[†] (Xu et al., 2021) | 72.79 | 500+ | 88.89 | 222 |
| FedDC[‡] (Gao et al., 2022) | 80.11 | **149** | 88.97 | 126 |
| FedACG (ours) | **80.61** | 169 | **90.09** | **108** |

## 4.4 EXPERIMENTS ON REALISTIC DATASETS

We conducted experiments on additional realistic datasets, FEMNIST and CelebA in LEAF (Caldas et al., 2019), which includes other non-iid scenarios such as feature skewness and data imbalance between clients. For the experiment, the number of clients is set to 2000 with data split following Caldas et al. (2019), and 10 randomly sampled clients participate the training for each communication round. We use simple CNN with group normalization with the number of layers two for FEMNIST and four for CelebA, respectively. Table 6 presents that FedACG also outperforms other baselines on both datasets for most cases, which supports our claim about the strength of FedACG on dataset heterogeneity. Table 6 presents that FedACG also outperforms other baselines on both datasets for most cases, which supports our claim about the strength of FedACG on dataset heterogeneity. Note that, while FedACG requires 20 more communication rounds than FedDC to reach a target accuracy on FEMNIST, it sends $1.76\times$ less parameters than FedDC.

## 5 CONCLUSION

This paper tackles a realistic federated learning scenario, where a large number of clients with heterogeneous data and limited participation constraints hurt the convergence and performance of the model. To address this problem, we proposed a novel federated learning framework, which naturally aggregates previous global gradient information and incorporates it to guide client updates. The proposed algorithm transmits the global gradient information to clients without additional communication cost by simply adding the global information to the current model when broadcasting it to clients. We showed that the proposed method is desirable with the realistic federated learning scenarios since it does not require any constraints such as communication or memory overhead. We demonstrate the effectiveness of the proposed method in terms of robustness and communication-efficiency in the presence of client heterogeneity through extensive evaluation on multiple benchmarks.

**Ethics statement** We propose a communication-efficient federated learning framework which handles non-*i.i.d* data distribution over remote clients. Without access to the raw data stored in remote devices, the proposed method gets the basic level of privacy. Also, unlike centralized training which suffers from dataset bias and unfairness problems since collected data reflects the perspec-

tive of the person who collects the data, it opens the way to learn real data distribution instead of collected data distribution.

**Reproducibility statement**    We present the procedure of our proposed method in Algorithm 1, and the implementation details in Section 4.1. We also present algorithm-dependent hyperparameters in Appendix B. We have submitted the code and will make it publicly available.

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

## A  CONVERGENCE OF FEDACG

We now present the theoretical convergence rate of FedACG. We first state few assumptions for the local loss functions $\ell_i(\theta)$, which are commonly used in several previous works on federated optimization (Karimireddy et al., 2020; Reddi et al., 2021; Xu et al., 2021). First, the local function $\mathcal{L}_i(\cdot)$ is assumed to be $L$-smooth for all $i \in \{1, \ldots, N\}$, *i.e.*,

$$\|\nabla \mathcal{L}_i(x) - \nabla \mathcal{L}_i(y)\| \leq L\|x - y\| \quad \forall x, y. \tag{7}$$

if local functions $\{\mathcal{L}_i(\cdot)\}_{i=1}^N$ are convex, we additionally have

$$\forall x, \qquad \frac{1}{2LN} \sum_{i=1}^N \|\nabla \mathcal{L}_i(x) - \nabla \mathcal{L}_i(x_*)\|^2 \leq \mathcal{L}(x) - \mathcal{L}(x_*) \text{ and} \tag{8}$$

$$\forall x, y, z, \ \langle \nabla \mathcal{L}_i(x), z - y \rangle \leq -\mathcal{L}_i(z) + \mathcal{L}_i(y) + \frac{L}{2}\|z - x\|^2, \tag{9}$$

where $\mathcal{L}(x) = \frac{1}{N} \sum_{i=1}^N \mathcal{L}_i(x)$ and $\nabla \mathcal{L}(x_*) = 0$. Second, we assume the local loss functions $\mathcal{L}_i(x)$ have bounded variance, *i.e.*, $\mathbb{E}_{\mathcal{D}_i}\|\nabla \ell_i(x) - \nabla \mathcal{L}_i(x)\| < \sigma^2$, and bounded gradients *i.e.*, $\|\mathcal{L}_i(x)\|_2 < G$, for all $x$. Based on the above assumptions, we derive the following asymptotic convergence bound of FedACG.

### A.1  NON-CONVEX ANALYSIS

**Theorem 1.** *(Convergence of FedACG) Suppose that local functions $\{f_i\}_{i=1}^N$ are non-convex and $L$-smooth. Let $z^t = \theta^t + \frac{\lambda}{1-\lambda}m^t$ for any $0 \leq \lambda < 1$. Then, by setting $\eta = \min\left\{\frac{1-\lambda}{2LK}, \frac{C}{\sqrt{t+1}}\right\}$, FedACG satisfies,*

$$\min_{k=0,\ldots,t} \mathbb{E}\left\|\nabla \mathcal{F}\left(\theta^t + \lambda m^t\right)\right\|^2 \leq \frac{2\left(\mathcal{F}(z_0) - \mathcal{F}_*\right)(1-\lambda)}{t+1} \max\left\{\frac{2LK}{1-\lambda}, \frac{\sqrt{t+1}}{C}\right\} + \frac{C}{\sqrt{t+1}}B'', \tag{10}$$

*where* $B'' = \frac{1}{(1-\lambda)K}\left\{\left((1 + \frac{L^2K^4}{3})(1-\lambda) + \frac{\lambda^4 LK^2}{2(1-\lambda)^2} + (1 + \frac{4N}{|S_t|(N-1)}(1 - \frac{|S_t|}{N}))(LK^2 + \frac{L\lambda^4 K^2}{2(1-\lambda)^2}))G^2 + \frac{LK^2}{2}\left(2 + \frac{\lambda^4}{(1-\lambda^2)}\right)\sigma^2\right\}.$

*Proof.* We start the proof from the result in Lemma 1,

$$\mathbb{E}[\mathcal{F}(z^{t+1}) - \mathcal{F}(z^{t+1})] \le -B\mathbb{E}[\|\nabla \mathcal{F}(\theta^t + \lambda m^t)\|^2] + B',$$

where $B = \frac{\eta K}{1-\lambda}\left(1 - \frac{\eta LK}{1-\lambda}\right)$, and $B' = \frac{\eta^2}{2(1-\lambda)^2}\Big\{\big((1+\frac{L^2 K^4}{3})(1-\lambda) + \frac{\lambda^4 LK^2}{2(1-\lambda)^2} + (1+\frac{4N}{|S_t|(N-1)}(1-\frac{|S_t|}{N}))(LK^2 + \frac{L\lambda^4 K^2}{2(1-\lambda)^2})\big)G^2 + \frac{LK^2}{2}\big(2 + \frac{\lambda^4}{(1-\lambda^2)}\big)\sigma^2\Big\}$, respectively.

By summing the above inequalities for $t = 0, \ldots, t$ and by noting that $\lambda < \frac{1-\lambda}{LK}$,

$$B\sum_{k=0}^{t}\mathbb{E}\|\nabla \mathcal{F}(\theta^t + \lambda m^t)\|^2 \le \mathbb{E}[\mathcal{F}(z^0) - \mathcal{F}(z^{t+1})] + (t+1)B'$$

$$\le \mathbb{E}[\mathcal{F}(z^0) - \mathcal{F}_*] + (t+1)B'.$$

Then

$$\min_{k=0,\ldots,t}\mathbb{E}\left\|\nabla \mathcal{F}\left(\theta^t + \lambda m^t\right)\right\|^2 \le \frac{f(\mathbf{z}_0) - f_*}{(t+1)B} + \frac{B'}{B} \tag{11}$$

Assume $\eta \le \frac{1-\lambda}{2LK}$, then $B \ge \frac{\eta K}{2(1-\lambda)}$. Then

$$\min_{k=0,\ldots,t}\mathbb{E}\left\|\nabla \mathcal{F}\left(\theta^t + \lambda m^t\right)\right\|^2 \le \frac{2(\mathcal{F}(z_0) - \mathcal{F}_*)(1-\lambda)}{\eta K(t+1)} + \frac{2(1-\lambda)}{\eta K}B'. \tag{12}$$

Noting that $\eta = \min\left\{\frac{1-\lambda}{2LK}, \frac{C}{\sqrt{t+1}}\right\}$, we can have

$$\min_{k=0,\ldots,t}\mathbb{E}\left\|\nabla \mathcal{F}\left(\theta^t + \lambda m^t\right)\right\|^2 \le \frac{2(\mathcal{F}(z_0) - \mathcal{F}_*)(1-\lambda)}{t+1}\max\left\{\frac{2LK}{1-\lambda}, \frac{\sqrt{t+1}}{C}\right\} + \frac{C}{\sqrt{t+1}}B'',$$
$$\tag{13}$$

where $B'' = \frac{1}{(1-\lambda)K}\Big\{\big((1+\frac{L^2 K^4}{3})(1-\lambda) + \frac{\lambda^4 LK^2}{2(1-\lambda)^2} + (1 + \frac{4N}{|S_t|(N-1)}(1-\frac{|S_t|}{N}))(LK^2 + \frac{L\lambda^4 K^2}{2(1-\lambda)^2})\big)G^2 + \frac{LK^2}{2}\big(2 + \frac{\lambda^4}{(1-\lambda^2)}\big)\sigma^2\Big\}$. We then complete the proof by noting that $z^0 = \theta^0 + \lambda m^0$. □

**Lemma 1.** *For proving Theorem 1, we first prove the key Lemma below. Let $z^t = \theta^t + \frac{\lambda}{1-\lambda}m^t$, $\Delta_i^t = \theta_i^t - (\theta^{t-1} + \lambda m^{t-1}) = \sum_{k=0}^{K-1}-\eta\nabla f_i(\theta_{i,k}^t)$, $\delta^t = \frac{1}{N}\sum_{i\in[N]}\sum_{k=0}^{K-1}-\eta\nabla \mathcal{F}_i(\theta_{i,k}^t)$, and $e^t = \Delta^t - \delta^t$. FedACG satisfies for any $t \ge 0$ and $0 \le \lambda < 1$,*

$$\mathrm{E}\left[f(\mathbf{z}_{k+1}) - f(\mathbf{z}_k)\right] \le -B\mathrm{E}\left[\|\nabla f(\mathbf{x}_k)\|^2\right] + B',$$

*where $B = \frac{\eta K}{1-\lambda}\left(1 - \frac{\eta LK}{1-\lambda}\right)$ and $B' = \frac{\eta^2}{2(1-\lambda)^2}\Big\{\big((1+\frac{L^2 K^4}{3})(1-\lambda) + \frac{\lambda^4 LK^2}{2(1-\lambda)^2} + (1 + \frac{4N}{|S_t|(N-1)}(1-\frac{|S_t|}{N}))(LK^2 + \frac{L\lambda^4 K^2}{2(1-\lambda)^2})\big)G^2 + \frac{LK^2}{2}\big(2 + \frac{\lambda^4}{(1-\lambda^2)}\big)\sigma^2\Big\}$*

*Proof.*

$$\mathcal{F}(z^{t+1}) \leq \mathcal{F}(z^t) + \langle \nabla \mathcal{F}(z^t), z^{t+1} - z^t \rangle + \frac{L}{2}\|z^{t+1} - z^t\|^2$$

$$= \mathcal{F}(z^t) + \frac{1}{1-\lambda}\langle \nabla \mathcal{F}(z^t), \Delta^{t+1} \rangle + \frac{L}{2(1-\lambda)^2}\|\Delta^{t+1}\|^2$$

$$= \mathcal{F}(z^t) + \frac{1}{1-\lambda}\langle \nabla \mathcal{F}(z^t), (e^{t+1} + \delta^{t+1}) + \eta K \nabla \mathcal{F}(\theta^t + \lambda m^t) - \eta K \nabla \mathcal{F}(\theta^t + \lambda m^t) \rangle$$

$$+ \frac{L}{2(1-\lambda)^2}\|e^{t+1} + \delta^{t+1}\|^2$$

$$= \mathcal{F}(z^t) + \frac{1}{1-\lambda}\langle \nabla \mathcal{F}(z^t), e^{t+1} \rangle + \frac{1}{1-\lambda}\langle \nabla \mathcal{F}(z^t), (\delta^{t+1} + \eta K \nabla \mathcal{F}(\theta^t + \lambda m^t)) \rangle$$

$$- \frac{\eta K}{1-\lambda}\langle \nabla \mathcal{F}(z^t), \nabla \mathcal{F}(\theta^t + \lambda m^t) \rangle + \frac{L}{2(1-\lambda)^2}\|e^{t+1} + \delta^{t+1}\|^2$$

$$= \mathcal{F}(z^t) + \frac{1}{1-\lambda}\langle \nabla \mathcal{F}(z^t), e^{t+1} \rangle + \frac{1}{1-\lambda}\langle \nabla \mathcal{F}(z^t), \delta^t + \eta K \nabla \mathcal{F}(\theta^t + \lambda m^t) \rangle$$

$$- \frac{\eta K}{1-\lambda}\langle \nabla \mathcal{F}(z^t) - \nabla \mathcal{F}(\theta^t + \lambda m^t), \nabla \mathcal{F}(\theta^t + \lambda m^t) \rangle - \frac{\eta K}{1-\lambda}\|\nabla \mathcal{F}(\theta^t + \lambda m^t)\|^2$$

$$+ \frac{L}{2(1-\lambda)^2}\|e^{t+1} + \delta^{t+1}\|^2$$

First inequality comes from the $L$-smoothness of the loss function $\mathcal{F}$. By taking expectation on both sides, we get following equation.

$$\mathbb{E}(\mathcal{F}(z^{t+1}) - \mathcal{F}(z^t)) \leq \frac{1}{1-\lambda}\mathbb{E}[\langle \nabla \mathcal{F}(z^t), \delta^t + \eta K \nabla \mathcal{F}(\theta^t + \lambda m^t) \rangle]$$

$$- \frac{\eta K}{1-\lambda}\mathbb{E}[\langle \nabla \mathcal{F}(z^t) - \nabla \mathcal{F}(\theta^t + \lambda m^t), \nabla \mathcal{F}(\theta^t + \lambda m^t) \rangle]$$

$$- \frac{\eta K}{1-\lambda}\mathbb{E}[\|\nabla \mathcal{F}(\theta^t + \lambda m^t)\|^2] + \frac{L}{2(1-\lambda)^2}\mathbb{E}[\|e^{t+1} + \delta^{t+1}\|^2]$$

$$\leq \frac{1}{2(1-\lambda)}\{\underbrace{\mathbb{E}[\|\eta \nabla \mathcal{F}(z^t)\|^2]}_{\text{I*}} + \underbrace{\mathbb{E}[\|\frac{1}{\eta}(\delta^t + \eta K \nabla \mathcal{F}(\theta^t + \lambda m^t))\|^2]}_{\text{II*}}\}$$

$$+ \frac{1}{4L}\underbrace{\mathbb{E}[\|\nabla \mathcal{F}(z^t) - \nabla \mathcal{F}(\theta^t + \lambda m^t)\|^2]}_{\text{III*}} + \frac{L}{2(1-\lambda)^2}(\underbrace{\mathbb{E}[\|e^{t+1}\|^2]}_{\text{IV*}} + \underbrace{\mathbb{E}[\|\delta^{t+1}\|^2]}_{\text{V*}})$$

$$+ \{L(\frac{K\eta}{1-\lambda})^2 - \frac{K\eta}{1-\lambda}\}\mathbb{E}[\|\nabla \mathcal{F}(\theta^t + \lambda m^t)\|^2]$$

First line holds because $\mathbb{E}[e^{t+1}] = 0$. Second inequality comes from the Lemma 5. Now we have to find the upper bound of five terms depicted in the last inequality. I*'s upper bound is,

$$\text{I*} = \eta^2 \mathbb{E}[\|\frac{1}{N}\sum_{i \in [N]} \nabla \mathcal{F}_i(z^t)\|^2] \leq \eta^2 G^2$$

The upper bound of II*, III*, and IV* are handled in Lemma 2, Lemma 3, and Lemma 4, respectively. V*'s upper bound is,

$$\text{V*} = \mathbb{E}[\|\frac{1}{N}\sum_{i \in [N]}\sum_{k=0}^{K-1} -\eta \nabla \mathcal{F}_i(\theta_{i,k}^t)\|^2] \leq \frac{\eta^2 K}{N}\sum_{i \in [N]}\sum_{k=0}^{K-1}\mathbb{E}[\|\nabla \mathcal{F}_i(\theta_{i,k}^t)\|^2] \leq \eta^2 K^2 G^2$$

Substituting upper bound for five items yields the desired result. $\square$

**Lemma 2.** $\mathbb{E}[\|\frac{1}{\eta}(\delta^t + \eta K \nabla \mathcal{F}(\theta^t + \lambda m^t))\|^2]$ *in the proof of Lemma 6 has following bound.*

$$\mathbb{E}[\|\frac{1}{\eta}(\delta^t + \eta K \nabla \mathcal{F}(\theta^t + \lambda m^t))\|^2] \leq \frac{\eta^2 L^2 K^4 G^2}{3}$$

*Proof.*

$$\mathbb{E}[\|\frac{1}{\eta}(\delta^t + \eta K \nabla \mathcal{F}(\theta^t + \lambda m^t))\|^2\} = \mathbb{E}[\|\frac{1}{\eta}(\delta^t + \eta K \nabla \mathcal{F}(\theta^t + \lambda m^t))\|^2$$

$$= \mathbb{E}[\|\frac{1}{N} \sum_{i \in [N]} \sum_{k=0}^{K-1} \{-\nabla \mathcal{F}_i(\theta_{i,k}^t) + \nabla \mathcal{F}_i(\theta^t + \lambda m^t))\}\|^2$$

$$\leq \frac{K}{N} \sum_{i \in [N]} \sum_{k=0}^{K-1} \mathbb{E}[\|\{-\nabla \mathcal{F}_i(\theta_{i,k}^t) + \nabla \mathcal{F}_i(\theta^t + \lambda m^t))\}\|^2$$

$$\leq \frac{L^2 K}{N} \sum_{i \in [N]} \sum_{k=0}^{K-1} \mathbb{E}[\|\theta_{i,k}^t - \theta_{i,0}^t\|^2$$

$$= \frac{L^2 K}{N} \sum_{i \in [N]} \sum_{k=0}^{K-1} \mathbb{E}[\|\sum_{\tau=0}^{k-1} -\eta \nabla \mathcal{F}_i(\theta_{i,k}^t)\|^2$$

$$\leq \frac{\eta^2 L^2 K}{N} \sum_{i \in [N]} \sum_{k=0}^{K-1} k \sum_{\tau=0}^{k-1} \mathbb{E}[\|\nabla \mathcal{F}_i(\theta_{i,k}^t)\|^2$$

$$\leq \frac{\eta^2 L^2 K}{N} \sum_{i \in [N]} \sum_{k=0}^{K-1} k^2 G^2$$

$$\leq \frac{\eta^2 L^2 K^4 G^2}{3}$$

Inequality in the third and sixth line comes from Jensen's inequality. Inequality in the fourth line is derived by the smoothness of the objective function. Inequality in the seventh line use bounded gradient assumption.

$\square$

**Lemma 3.** $\mathbb{E}\|\nabla \mathcal{F}(z^t) - \nabla \mathcal{F}(\theta^t + \lambda m^t)\|^2$ *in the proof of Lemma 6 has the following bound for any* $0 \leq \lambda < 1$,

$$\mathbb{E}\|\nabla \mathcal{F}(z^t) - \nabla \mathcal{F}(\theta^t + \lambda m^t)\|^2 \leq \frac{\eta^2 \lambda^4 L^2 K^2 G^2 \left\{1 + \frac{4N}{|\mathcal{S}_t|(N-1)}\left(1 - \frac{|\mathcal{S}_t|}{N}\right)\right\} + \eta^2 \lambda^4 L^2 K^2 \sigma^2}{(1-\lambda)^4}$$

*Proof.*

$$\mathbb{E}\|\nabla \mathcal{F}(z^t) - \nabla \mathcal{F}(\theta^t + \lambda m^t)\|^2 \leq L^2 \mathbb{E}\|\frac{\lambda^2}{1-\lambda} m^t\|^2 = \frac{\lambda^4 L^2}{(1-\lambda)^2} \mathbb{E}\|m^t\|^2$$

$$= \frac{\lambda^4 L^2}{(1-\lambda)^2} \mathbb{E}\|\sum_{k=0}^{t} \lambda^{t-k} \Delta^k\|^2,$$

where the first inequality comes from the $L$-smoothness of the global loss function $\mathcal{F}$, while the last equation comes from the unrolling the recursion of the momentum $m^t$, *i.e.*, $m^t = \sum_{k=0}^{t} \lambda^{t-k} \Delta^k$. Let $\Gamma_t = \sum_{k=0}^{t} \lambda^k = \frac{1-\lambda^t}{1-\lambda}$. For $0 \leq \lambda < 1, \Gamma_t \leq \frac{1}{1-\lambda}$. Then

$$\frac{\lambda^4 L^2}{(1-\lambda)^2} \mathbb{E}\|\sum_{k=0}^{t} \lambda^{t-k} \Delta^k\|^2 = \frac{\lambda^4 L^2}{(1-\lambda)^2} \Gamma_t^2 \|\frac{1}{\Gamma_t} \sum_{k=1}^{t} \lambda^{t-k} \Delta^k\|^2$$

$$\leq \frac{\lambda^4 L^2}{(1-\lambda)^2} \Gamma_t \sum_{k=1}^{t} \lambda^{t-k} \mathbb{E}\|\Delta^k\|^2 = \frac{\lambda^4 L^2}{(1-\lambda)^2} \Gamma_t^2 \mathbb{E}\|\Delta^t\|^2$$

$$= \frac{\eta^2 \lambda^4 L^2 K^2 G^2 \left\{1 + \frac{4N}{|\mathcal{S}_t|(N-1)}\left(1 - \frac{|\mathcal{S}_t|}{N}\right)\right\} + \eta^2 \lambda^4 L^2 K^2 \sigma^2}{(1-\lambda)^4}$$

$\square$

**Lemma 4.** $\mathbb{E}\|e\|^2$ *in the proof of Lemma 6 has the following bound,*

$$\mathbb{E}\|e^t\|^2 \leq \eta^2 K^2 \sigma^2 + \frac{4\eta^2 N K^2}{(N-1)|\mathcal{S}_t|}(1 - \frac{|\mathcal{S}_t|}{N})G^2$$

*Proof.* We have $\mathbb{E}\|e^t\|^2 = \mathbb{E}\|\Delta^t - \delta^t\|$. Note that:

$$\mathbb{E}\|\Delta^t - \delta^t\|^2 = \mathbb{E}\|\Delta^t + \frac{\eta}{|\mathcal{S}_t|}\sum_{i\in\mathcal{S}_t}\sum_{k=0}^{K-1}\nabla\mathcal{F}_i(\theta_{i,k}^t) - \frac{\eta}{|\mathcal{S}_t|}\sum_{i\in\mathcal{S}_t}\sum_{k=0}^{K-1}\nabla\mathcal{F}_i(\theta_{i,k}^t) + \frac{\eta}{N}\sum_{i\in[N]}\sum_{k=0}^{K-1}\nabla\mathcal{F}_i(\theta_{i,k}^t)\|^2$$

$$= \mathbb{E}\|\frac{\eta}{|\mathcal{S}_t|}\sum_{i\in\mathcal{S}_t}\sum_{k=0}^{K-1}\nabla f_i(\theta_{i,k}^t) - \nabla\mathcal{F}_i(\theta_{i,k}^t)\|^2 + \mathbb{E}\|\frac{\eta}{|\mathcal{S}_t|}\sum_{i\in\mathcal{S}_t}\sum_{k=0}^{K-1}\nabla\mathcal{F}_i(\theta_{i,k}^t) - \delta^t\|^2$$

$$\leq \frac{\eta^2}{|\mathcal{S}_t|}\sum_{i\in\mathcal{S}_t}\sum_{k=0}^{K-1}\mathbb{E}\|\nabla f_i(\theta_{i,k}^t) - \nabla\mathcal{F}_i(\theta_{i,k}^t)\|^2 + \mathbb{E}\|\frac{\eta}{|\mathcal{S}_t|}\sum_{i\in\mathcal{S}_t}\sum_{k=0}^{K-1}\nabla\mathcal{F}_i(\theta_{i,k}^t) - \delta^t\|^2$$

$$= \eta^2 K^2 \sigma^2 + \underbrace{\mathbb{E}\|\frac{\eta}{|\mathcal{S}_t|}\sum_{i\in\mathcal{S}_t}\sum_{k=0}^{K-1}\nabla\mathcal{F}_i(\theta_{i,k}^t) - \delta^t\|^2}_{(A)}$$

In (A), we take expectation with respect to $\mathcal{S}_k$ and total clients $N$. For that, we use Lemma 4 of Reisizadeh et al. (2020). Specifically, using Eq. (59) in Reisizadeh et al. (2020), we get:

$$(A) \leq \frac{\eta^2}{|\mathcal{S}_t|^2}\left(\frac{|\mathcal{S}_t|}{N} - \frac{|\mathcal{S}_t|(|\mathcal{S}_t|-1)}{N(N-1)}\right)\sum_{i\in[N]}\mathbb{E}\|\sum_{k=0}^{K-1}\nabla\mathcal{F}_i(\theta_{i,k}^t) - \frac{\delta^t}{\eta}\|^2$$

$$\leq \frac{\eta^2}{|\mathcal{S}_t|^2}\left(\frac{|\mathcal{S}_t|}{N} - \frac{|\mathcal{S}_t|(|\mathcal{S}_t|-1)}{N(N-1)}\right)\left(2\sum_{i\in[N]}\mathbb{E}\|\sum_{k=0}^{K-1}\nabla\mathcal{F}_i(\theta)\|^2 + \frac{2}{N}\sum_{i\in[N]}\mathbb{E}\|\sum_{k=0}^{K-1}\nabla\mathcal{F}_i(\theta_{i,k}^t)\|^2\right)$$

$$\leq \frac{4\eta^2}{|\mathcal{S}_t|^2}\left(\frac{|\mathcal{S}_t|}{N} - \frac{|\mathcal{S}_t|(|\mathcal{S}_t|-1)}{N(N-1)}\right)\sum_{i\in[N]}\mathbb{E}\|\sum_{k=0}^{K-1}\nabla\mathcal{F}_i(\theta_{i,k}^t)\|^2$$

$$\leq \frac{4\eta^2}{|\mathcal{S}_t|^2}\left(\frac{|\mathcal{S}_t|}{N} - \frac{|\mathcal{S}_t|(|\mathcal{S}_t|-1)}{N(N-1)}\right)NK^2G^2$$

This gives us the desired result. $\square$

**Lemma 5.** *(Relaxed triangle inequality).* *For any* $a > 0$, $\|\boldsymbol{v}_1 + \boldsymbol{v}_2\|^2 \leq (1+a)\|\boldsymbol{v}_1\|^2 + \left(1 + \frac{1}{a}\right)\|\boldsymbol{v}_2\|^2$

*Proof.* This lemma holds because when we organize the formulas on the right, we get $0 \leq \|a\boldsymbol{v}_1 - \frac{\boldsymbol{v}_2}{a}\|^2$. $\square$

### A.2 CONVEX ANALYSIS

**Theorem 2.** *Suppose that local functions* $\{\ell_i\}_{i=1}^N$ *are convex and L-smooth. Then, for* $\frac{1}{2} < \lambda < 1$ *and* $\frac{L}{1-\lambda} \leq \beta$, *FedACG satisfies,*

$$\mathbb{E}\left[\ell\left(\frac{1}{T}\sum_{t=1}^T\theta^{t-1}\right) - \ell(\theta_*)\right] \leq \frac{1}{T}\left(\beta(1-\lambda)\|\theta^0 - \theta_*\|^2 + \frac{\lambda}{\beta}\left(\frac{1}{N}\sum_{i=1}^N\|\nabla\ell_i(\theta_i^0)\|^2\right)\right)$$

$$= O\left(\frac{1}{T}\right)$$

*where* $\theta_* = \underset{\theta}{\operatorname{argmin}}\,\ell(\theta)$, $\theta^t = \frac{1}{|S_t|}\sum_{i\in S_t}\theta_i^t$.

If $\beta = L\sqrt{\frac{\lambda}{1-\lambda}}$, we get the statement in Theorem 1 in the main paper. We utilize similar approaches as in SCAFFOLD (Karimireddy et al., 2020) and FedDyn (Acar et al., 2021) analysis throughout the proof. We define momentum $m^t = \theta^t - \theta^{t-1}$, and a set of variables for the analysis. Following analysis in FedDyn (Acar et al., 2021), we first define virtual variable $\{\tilde{\theta}_i^t\}$ as,

$$\tilde{\theta}_i^t = \underset{\theta}{\text{argmin}}\, \ell_i(\theta) + \frac{\beta}{2}\|\theta - (\theta^{t-1} + \lambda m^{t-1})\|^2. \tag{14}$$

$\theta^t$ consists of the locally trained models from participating devices. We express the server model as active device average and its relation with accelerated model as,

$$\theta^t = \frac{1}{|S_t|}\sum_{i \in S_t} \theta_i^t; \quad \theta^t = \gamma^t - \lambda m^t. \tag{15}$$

We also define $\epsilon_t$ which calculate difference between local models and the average of device models from previous round as,

$$\epsilon_t = \frac{1}{N}\sum_{i \in \{1,\dots,N\}} \mathbb{E}\|\tilde{\theta}_i^t - \theta^{t-1}\|^2. \tag{16}$$

If models converge to $\theta_*$, $\epsilon_t$ will be 0. After these definitions, Theorem 2 can be seen as a direct consequence of the following Lemma,

**Lemma 6.** *For convex and L-smooth $\{f_i\}_{i=1}^N$ functions, if $\frac{1}{2} < \lambda < 1$ and $\frac{L}{1-\lambda} \le \beta$, FedACG satisfies*

$$\mathbb{E}\|\theta^t - \theta_*\|^2 + \kappa\epsilon_t \le \mathbb{E}\|\theta^{t-1} - \theta_*\|^2 + \kappa\epsilon_{t-1} - \kappa_0 \mathbb{E}\left[\ell(\theta^{t-1}) - \ell(\theta_*)\right]$$

*where $\theta_* = \underset{\theta}{\text{argmin}}\, f(\theta)$, $\kappa = \frac{4\beta\lambda^2(L-\beta-\beta\lambda)}{(\beta^2 - 4L^2 - 4\beta^2\lambda^2)(1-\lambda)}$, $\kappa_0 = \frac{2}{\beta(1-\lambda)}\left(\frac{4L(L-\beta-\lambda\beta)}{\beta^2 - 4L^2 - 4\beta^2\lambda^2} - 1\right)$.*

Lemma 6 can be telescoped in the following way,

$$\kappa_0 \mathbb{E}\left[\ell(\theta^{t-1}) - \ell(\theta_*)\right] \le \left(\mathbb{E}\|\theta^{t-1} - \theta_*\|^2 + \kappa\epsilon_{t-1}\right) - \left(\mathbb{E}\|\theta^t - \theta_*\|^2 + \kappa\epsilon_t\right)$$

$$\kappa_0 \sum_{t=1}^T \mathbb{E}\left[\ell(\theta^{t-1}) - \ell(\theta_*)\right] \le \left(\mathbb{E}\|\theta^0 - \theta_*\|^2 + \kappa\epsilon_0\right) - \left(\mathbb{E}\|\theta^T - \theta_*\|^2 + \kappa\epsilon_T\right)$$

If $\frac{1}{2} < \lambda < 1$ and $\frac{L}{1-\lambda} \le \beta$, $\kappa_0$ and $\kappa$ become positive. Eliminating negative terms on RHS gives,

$$\kappa_0 \sum_{t=1}^T \mathbb{E}\left[\ell(\theta^{t-1}) - \ell(\theta_*)\right] \le \mathbb{E}\|\theta^0 - \theta_*\|^2 + \kappa\epsilon_0$$

Applying Jensen's inequality on LHS gives,

$$\mathbb{E}\left[\ell\left(\frac{1}{T}\sum_{t=1}^T \theta^{t-1}\right) - \ell(\theta_*)\right] \le \frac{1}{T}\frac{1}{\kappa_0}\left(\|\theta^0 - \theta_*\|^2 + \kappa\epsilon_0\right) = O\left(\frac{1}{T}\right),$$

which proves the statement in Theorem 2. Similar to convergence analysis of gradient descent, $\|\theta^t - \theta_*\|^2$ is expressed as $\|\theta^t - \theta^{t-1} + \theta^{t-1} - \theta_*\|^2$ and expanded in the proof of Lemma 6. To tackle the extra terms, we state the following Lemmas and corresponding proofs. We first bound $\|\theta^t - \theta^{t-1}\|^2$ with the following,

**Lemma 7.** *Suppose that local functions $\{f_i\}_{i=1}^N$ are convex and L-smooth. Then we can bound the global model update,*

$$E\|\theta^t - \theta^{t-1}\|^2 \le \epsilon_t \tag{17}$$

*Proof.* Note that

$$\mathbb{E}\|\theta^t - \theta^{t-1}\|^2 = \mathbb{E}\left\|\frac{1}{|S_t|}\sum_{i \in \mathcal{S}_t}\left(\theta_i^t - \theta^{t-1}\right)\right\|^2 \le \frac{1}{|S_t|}\mathbb{E}\left[\sum_{i \in \mathcal{S}_t}\|\theta_i^t - \theta^{t-1}\|^2\right]$$

$$= \frac{1}{|S_t|}\mathbb{E}\left[\sum_{i \in \mathcal{S}_t}\left\|\tilde{\theta}_i^t - \theta^{t-1}\right\|^2\right] = \frac{1}{|S_t|}\frac{|S_t|}{N}\sum_{i=1}^N \mathbb{E}\left\|\tilde{\theta}_i^t - \theta^{t-1}\right\|^2$$

$$= \epsilon_t$$

$\square$

where first equality comes from Eq. (15). The following inequality applies Jensen. Remaining relations are due to $\tilde{\theta}_i^t = \theta_i^t$ if $i \in \mathcal{S}_t$, taking expectation by conditioning on randomness before time $t$ and definition of $\epsilon_t$.

We introduce additional Lemma to further bound $\epsilon$ term in Lemma 7. Before the proof, we first introduce triangular inequality here.

**Lemma 8.** $\forall \{v_j\}_{j=1}^n \in \mathcal{R}^d$, triangular inequality satisfies

$$\left\| \sum_{j=1}^n v_j \right\|^2 \leq n \sum_{j=1}^n \|v_j\|^2.$$

*Proof.* With Jensen's inequality, $\left\| \frac{1}{n} \sum_{j=1}^n v_j \right\|^2 \leq \frac{1}{n} \sum_{j=1}^n \|v_j\|^2$. Multiplying both sides with $n^2$ gives the inequality.

**Lemma 9.** For convex and L-smooth $\{f_i\}_{i=1}^N$ functions, then the updates of FedACG have bounded drift,

$$\left(1 - \frac{4L^2}{\beta^2}\right) \epsilon_t \leq 4\lambda^2 \epsilon_{t-1} + \frac{8L}{\beta^2} \mathbb{E}\left[\ell(\theta^{t-1}) - \ell(\theta_*)\right] \tag{18}$$

*Proof.*

$$\epsilon_t = \frac{1}{N} \sum_{i=1}^N \mathbb{E}\|\tilde{\theta}_i^t - \theta^{t-1}\|^2$$

$$= \frac{1}{N} \sum_{i=1}^N \mathbb{E}\left\| -\frac{1}{\beta}\nabla\ell_i(\tilde{\theta}_i^t) + \lambda m^{t-1} \right\|^2$$

$$= \frac{1}{N} \sum_{i=1}^N \mathbb{E}\left\| -\frac{1}{\beta}\nabla\ell_i(\theta_*) + \frac{1}{\beta}\nabla\ell_i(\theta_*) - \frac{1}{\beta}\nabla\ell_i(\theta^{t-1}) + \frac{1}{\beta}\nabla\ell_i(\theta^{t-1}) \right.$$

$$\left. - \frac{1}{\beta}\nabla\ell_i(\tilde{\theta}_i^t) + \lambda m^{t-1} \right\|^2$$

$$\leq \frac{4}{\beta^2} \frac{1}{N} \sum_{i=1}^N \mathbb{E}\|\nabla\ell_i(\theta^{t-1}) - \nabla\ell_i(\theta_*)\|^2 + \frac{4}{\beta^2} \frac{1}{N} \sum_{i=1}^N \mathbb{E}\|\nabla\ell_i(\theta_*)\|^2$$

$$+ \frac{4}{\beta^2} \frac{1}{N} \sum_{i=1}^N \mathbb{E}\|\nabla\ell_i(\tilde{\theta}_i^t) - \nabla\ell_i(\theta^{t-1})\|^2 + 4\lambda^2 \epsilon_{t-1}$$

$$\leq \frac{4L^2}{\beta^2} \epsilon_t + 4\lambda^2 \epsilon_{t-1} + \frac{8L}{\beta^2} \mathbb{E}\left[\ell(\theta^{t-1}) - \ell(\theta_*)\right]$$

where first and second equations come from Eq. (15) and first order condition of Eq. (14). Following inequalities come from Lemma 8, 7, and convexity.

Now, let's express $\|\theta^t - \theta_*\|^2$ term as,

$$
\begin{aligned}
\mathbb{E}\|\theta^t - \theta_*\|^2 &= \mathbb{E}\|\theta^{t-1} - \theta_* + \theta^t - \theta^{t-1}\|^2 \\
&= \mathbb{E}\|\theta^{t-1} - \theta_*\|^2 + 2\mathbb{E}\left[\left\langle \theta^{t-1} - \theta_*, \theta^t - \theta^{t-1}\right\rangle\right] + \mathbb{E}\|\theta^t - \theta^{t-1}\|^2 \\
&\approx \mathbb{E}\|\theta^{t-1} - \theta_*\|^2 + \frac{2}{(1-\lambda)\beta N}\sum_{i=1}^{N}\mathbb{E}\left[\left\langle \theta^{t-1} - \theta_*, -\nabla\ell_i(\tilde{\theta}_i^t)\right\rangle\right] \\
&\quad + \mathbb{E}\|\theta^t - \theta^{t-1}\|^2 \\
&\leq \mathbb{E}\|\theta^{t-1} - \theta_*\|^2 + \mathbb{E}\|\theta^t - \theta^{t-1}\|^2 \\
&\quad + \frac{2}{(1-\lambda)\beta N}\sum_{i=1}^{N}\mathbb{E}\left[\ell_i(\theta_*) - \ell_i(\theta^{t-1}) + \frac{L}{2}\|\tilde{\theta}_i^t - \theta^{t-1}\|^2\right] \\
&= \mathbb{E}\|\theta^{t-1} - \theta_*\|^2 - \frac{2}{(1-\lambda)\beta}\mathbb{E}\left[\ell(\theta^{t-1}) - \ell(\theta_*)\right] \\
&\quad + \frac{L}{(1-\lambda)\beta}\epsilon_t + \mathbb{E}\|\theta^t - \theta^{t-1}\|^2
\end{aligned}
\tag{19}
$$

where we approximately have $\mathbb{E}[m^t] \approx -\frac{1}{(1-\lambda)\beta N}\sum_{i=1}^{N}\mathbb{E}[\nabla\ell_i(\theta_i^t)]$ since global gradient information is an exponentially updated with a coefficient $\lambda$. Following inequality is due to the quadratic bound by convexity and $L$-smoothness of local functions.

Let's scale Eq. (19) with $\frac{(1-\lambda)(\beta^2 - 4L^2 - 4\lambda^2\beta^2)}{\beta(L - \beta - \beta\lambda)}$. Note that the coefficient is positive due to the condition on $\beta$ and $\lambda$. Summing scaled version of Eq. (19) and Lemma 9 gives the statement in Lemma 6.

## B  HYPERPARAMETER SETTING

For the hyperparameter selection, we assume the scenario that the server can compute the validation accuracy through communication with clients at the early stages, which is common to all algorithms.

For the experiments on CIFAR-10 and CIFAR-100, we choose 5 as the number of local training epochs (50 iterations) and 0.1 as the local learning rate. We set the batch size of the local update to 50 and 10 for the 100 and 500 client participation, respectively. The learning rate decay parameter of each algorithm is selected from $\{0.995, 0.998, 1\}$ to achieve the best performance. The global learning rate is set to 1, except for FedAdam, which is tested with 0.01.

For the experiments on Tiny-ImageNet, we match the total local iterations of local updates with other benchmarks by setting the batch size of local updates as 100 and 20 for the 100 and 500 client participation, respectively.

As for algorithm-dependent hyperparameters, $\alpha$ in FedCM is selected from $\{0.1, 0.3, 0.5\}$, $\alpha$ in FedDyn is selected from $\{0.001, 0.01, 0.1\}$, and $\alpha$ in FedDC is set to 0.01. $\tau$ in FedAdam is set to 0.001 while $\mu$ in MOON is set to 1. $\beta$ in FedAvgM is selected from $\{0.4, 0.6, 0.8\}$, $\beta$ in FedProx and FedACG is selected from $\{0.1, 0.01, 0.001\}$, and $\lambda$ in FedACG is selected from $\{0.8, 0.85, 0.9\}$.

We submitted and will release the source code to facilitate the reproduction of our results.

## C  EVALUATION ON VARIOUS DATA HETEROGENEITY

Tables 7 and 8 show that FedACG matches or outperforms the performance of competitive methods when data heterogeneity is not severe (Dirichlet 0.6) or absent (IID) on CIFAR-10 in most cases. Note that, while the compared methods show performance degradation as the participation rate decreases, FedACG shows little degradation as the participation rate decreases for both data splits. This implies that FedACG is more robust for low participation rates than other baselines. This is partly because low client heterogeneity reduces noise in the momentum of global gradient, which attributes to the smooth trajectory of global update. Since FedACG effectively incorporates the momentum for local updates, FedACG is relatively unaffected by the partial participation of federated learning.

Table 7: Results with Dirichlet (0.6) data split on CIFAR-10 and CIFAR-100 for two different federated learning settings. Accuracy at the target round and the communication round to reach target test accuracy are based on running exponential moving average with parameter 0.9. The arrows indicate whether higher ($\uparrow$) or lower ($\downarrow$) is better. FedCM[†] and FedDC[‡] require $1.5\times$ and $2\times$ communication cost for each communication round, respectively.

(a) Dirichlet (0.6), 100 clients, 5% participation

| Method | CIFAR-10 | | | | CIFAR-100 | | | |
|---|---|---|---|---|---|---|---|---|
| | acc. (%, $\uparrow$) | | rounds ($\downarrow$) | | acc. (%, $\uparrow$) | | rounds ($\downarrow$) | |
| | 500R | 1000R | 81% | 87% | 500R | 1000R | 50% | 56% |
| FedAvg (McMahan et al., 2017) | 80.56 | 85.97 | 520 | 1000+ | 43.91 | 49.18 | 1000+ | 1000+ |
| FedProx (Li et al., 2020) | 80.39 | 85.53 | 524 | 1000+ | 43.15 | 48.45 | 1000+ | 1000+ |
| FedAvgM (Hsu et al., 2019) | 84.65 | 87.96 | 355 | 811 | 46.66 | 52.49 | 735 | 1000+ |
| FedADAM (Reddi et al., 2021) | 80.25 | 83.52 | 526 | 1000+ | 45.95 | 51.63 | 778 | 1000+ |
| FedDyn (Acar et al., 2021) | 87.23 | 89.49 | 310 | 487 | 50.51 | 56.78 | 488 | 886 |
| MOON (Li et al., 2021) | 84.95 | 87.99 | 272 | 728 | 55.76 | 61.42 | 338 | 527 |
| FedCM[†] (Xu et al., 2021) | 82.84 | 86.64 | 385 | 1000+ | 53.75 | 60.48 | 331 | 468 |
| FedDC[‡] (Gao et al., 2022) | **88.05** | 89.58 | 270 | **437** | 56.00 | 60.58 | 347 | 491 |
| FedACG (ours) | 87.57 | **90.56** | **218** | 453 | **58.82** | **63.88** | **243** | **396** |

(b) Dirichlet (0.6), 500 clients, 2% participation

| Method | CIFAR-10 | | | | CIFAR-100 | | | |
|---|---|---|---|---|---|---|---|---|
| | acc. (%, $\uparrow$) | | rounds ($\downarrow$) | | acc. (%, $\uparrow$) | | rounds ($\downarrow$) | |
| | 500R | 1000R | 69% | 80% | 500R | 1000R | 32% | 41% |
| FedAvg (McMahan et al., 2017) | 62.79 | 75.17 | 671 | 1000+ | 29.41 | 36.62 | 648 | 1000+ |
| FedProx (Li et al., 2020) | 62.48 | 75.10 | 688 | 1000+ | 29.62 | 36.70 | 647 | 1000+ |
| FedAvgM (Hsu et al., 2019) | 69.10 | 80.26 | 498 | 981 | 32.78 | 41.93 | 468 | 942 |
| FedADAM (Reddi et al., 2021) | 68.48 | 78.92 | 535 | 1000+ | 37.57 | 48.29 | 341 | 624 |
| FedDyn (Acar et al., 2021) | 68.53 | 80.33 | 513 | 983 | 32.06 | 43.28 | 498 | 917 |
| MOON (Li et al., 2021) | 74.29 | 80.66 | 368 | 921 | 31.64 | 41.61 | 515 | 931 |
| FedCM[†] (Xu et al., 2021) | 71.42 | 78.94 | 429 | 1000+ | 26.82 | 39.78 | 714 | 1000+ |
| FedDC[‡] (Gao et al., 2022) | 77.74 | **86.32** | 324 | 596 | 34.24 | 44.69 | 444 | 825 |
| FedACG (ours) | **78.49** | 85.28 | **289** | **565** | **39.61** | **49.70** | **304** | **540** |

# D    CONVERGENCE PLOT

## D.1    EVALUATION ON VARIOUS FEDERATED LEARNING SCENARIOS

Figure 2 to Figure 4 show the convergence of FedACG and the compared algorithms on CIFAR-10, CIFAR-100, and Tiny-ImageNet for various federated learning settings: varying the number of total clients, participation rates, data heterogeneity. FedACG continuously matches or exceeds the performance of the most powerful of our competitors in most learning sections.

Figure 5 shows the convergence plots under massive clients, 1% participation rate setting. The result shows that FedACG takes the lead in most learning sections, which also demonstrates the effectiveness of FedACG.

## D.2    EVALUATION ON DYNAMIC CLIENT SET

Figure 6 shows a convergence plot when the entire client's pool changes during training. The result shows that FedACG outperforms the baselines in most learning sections. Note that FedDyn shows worse performance than FedAvg in the overall section of learning, and only achieves FedAvg's performance at the end. This is partly because it needs to store local states for local training in each client, which requires a kind of warm-up period for newly participating clients to contain useful information. In contrast, FedACG, which is free from these restrictions, shows strength in a realistic federated learning scenario where the pool of the entire clients changes during training.

Table 8: Results with IID data split on CIFAR-10 and CIFAR-100 for two different federated learning settings. Accuracy at the target round and the communication round to reach target test accuracy are based on running exponential moving average with parameter 0.9. The arrows indicate whether higher ($\uparrow$) or lower ($\downarrow$) is better. FedCM[†] and FedDC[‡] require $1.5\times$ and $2\times$ communication cost for each communication round, respectively.

(a) IID, 100 clients, 5% participation

| Method | CIFAR-10 | | | | CIFAR-100 | | | |
| --- | --- | --- | --- | --- | --- | --- | --- | --- |
| | acc. (%, $\uparrow$) | | rounds ($\downarrow$) | | acc. (%, $\uparrow$) | | rounds ($\downarrow$) | |
| | 500R | 1000R | 82% | 89% | 500R | 1000R | 52% | 58% |
| FedAvg (McMahan et al., 2017) | 85.28 | 88.69 | 372 | 1000+ | 43.96 | 48.20 | 1000+ | 1000+ |
| FedProx (Li et al., 2020) | 84.79 | 87.99 | 384 | 1000+ | 43.57 | 47.75 | 1000+ | 1000+ |
| FedAvgM (Hsu et al., 2019) | 87.67 | 89.96 | 258 | 375 | 47.43 | 52.83 | 880 | 1000+ |
| FedADAM (Reddi et al., 2021) | 85.29 | 87.97 | 286 | 1000+ | 52.23 | 57.73 | 496 | 1000+ |
| FedDyn (Acar et al., 2021) | 89.19 | 90.70 | 269 | 492 | 50.37 | 56.88 | 592 | 898 |
| MOON (Li et al., 2021) | 88.24 | 89.96 | 207 | 628 | 58.50 | **64.73** | 311 | 484 |
| FedCM[†] (Xu et al., 2021) | 87.38 | 89.65 | 182 | 782 | 57.10 | 62.48 | 266 | 466 |
| FedDC[‡] (Gao et al., 2022) | 90.07 | 90.80 | 194 | 425 | 55.17 | 61.00 | 400 | 633 |
| FedACG (ours) | **90.57** | **92.29** | **157** | **354** | **59.82** | 64.08 | **244** | **342** |

(b) IID, 500 clients, 2% participation

| Method | CIFAR-10 | | | | CIFAR-100 | | | |
| --- | --- | --- | --- | --- | --- | --- | --- | --- |
| | acc. (%, $\uparrow$) | | rounds ($\downarrow$) | | acc. (%, $\uparrow$) | | rounds ($\downarrow$) | |
| | 500R | 1000R | 75% | 83% | 500R | 1000R | 33% | 42% |
| FedAvg (McMahan et al., 2017) | 68.70 | 78.21 | 652 | 1000+ | 30.71 | 37.85 | 664 | 1000+ |
| FedProx (Li et al., 2020) | 68.74 | 77.96 | 643 | 1000+ | 30.11 | 37.13 | 685 | 1000+ |
| FedAvgM (Hsu et al., 2019) | 74.34 | 83.64 | 523 | 943 | 33.54 | 42.55 | 479 | 971 |
| FedADAM (Reddi et al., 2021) | 75.32 | 84.01 | 491 | 915 | 38.74 | 48.94 | 328 | 636 |
| FedDyn (Acar et al., 2021) | 74.81 | 84.71 | 398 | 823 | 33.20 | 42.91 | 492 | 936 |
| MOON (Li et al., 2021) | 69.86 | 81.89 | 586 | 1000+ | 28.82 | 41.26 | 649 | 1000+ |
| FedCM[†] (Xu et al., 2021) | 77.84 | 83.26 | 491 | 959 | 29.59 | 42.04 | 653 | 991 |
| FedDC[‡] (Gao et al., 2022) | **80.87** | **87.53** | 358 | **574** | 33.93 | 45.80 | 476 | 817 |
| FedACG (ours) | 80.15 | 87.47 | **316** | 578 | **41.16** | **49.10** | **299** | **525** |

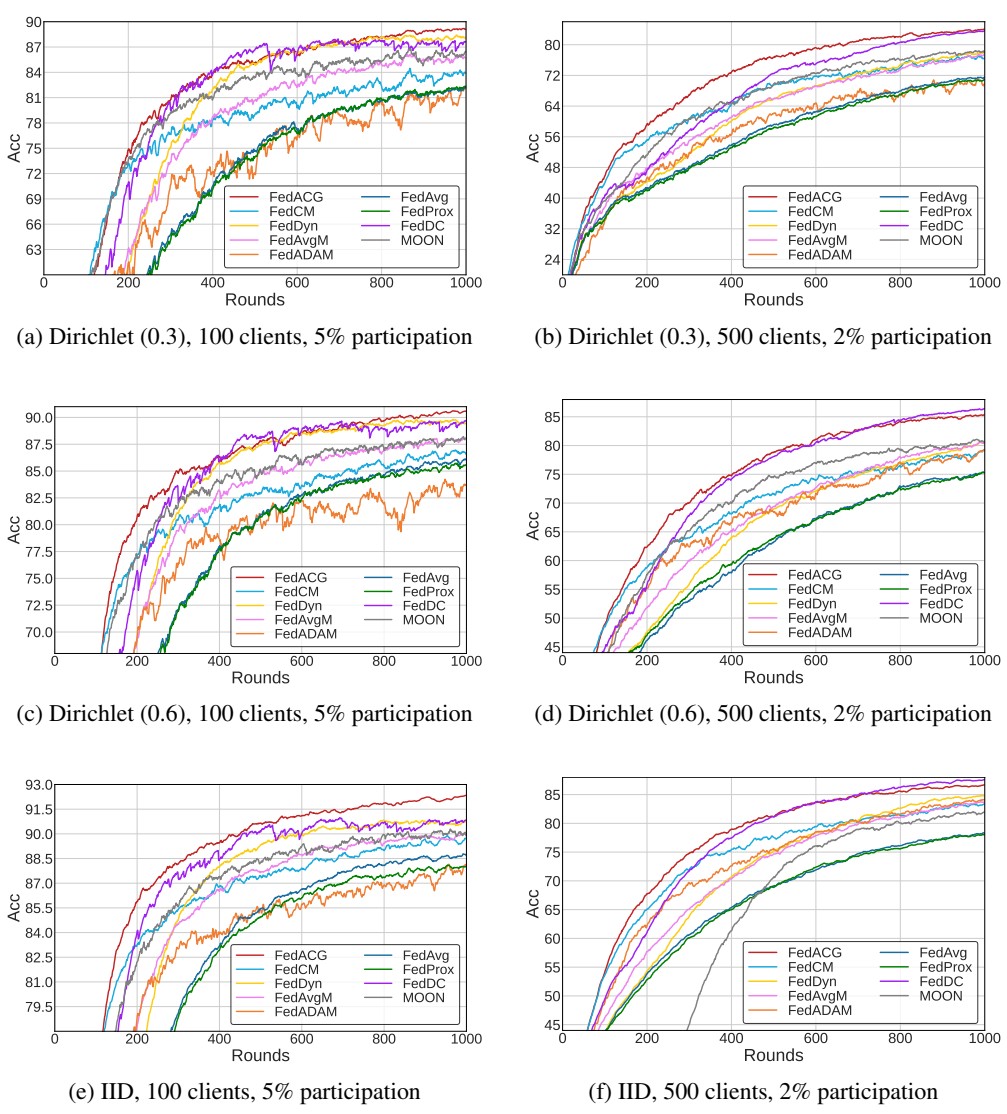

(a) Dirichlet (0.3), 100 clients, 5% participation

(b) Dirichlet (0.3), 500 clients, 2% participation

(c) Dirichlet (0.6), 100 clients, 5% participation

(d) Dirichlet (0.6), 500 clients, 2% participation

(e) IID, 100 clients, 5% participation

(f) IID, 500 clients, 2% participation

Figure 2: The convergence plots of FedACG and the baselines on CIFAR-10 with different federated learning scenarios.

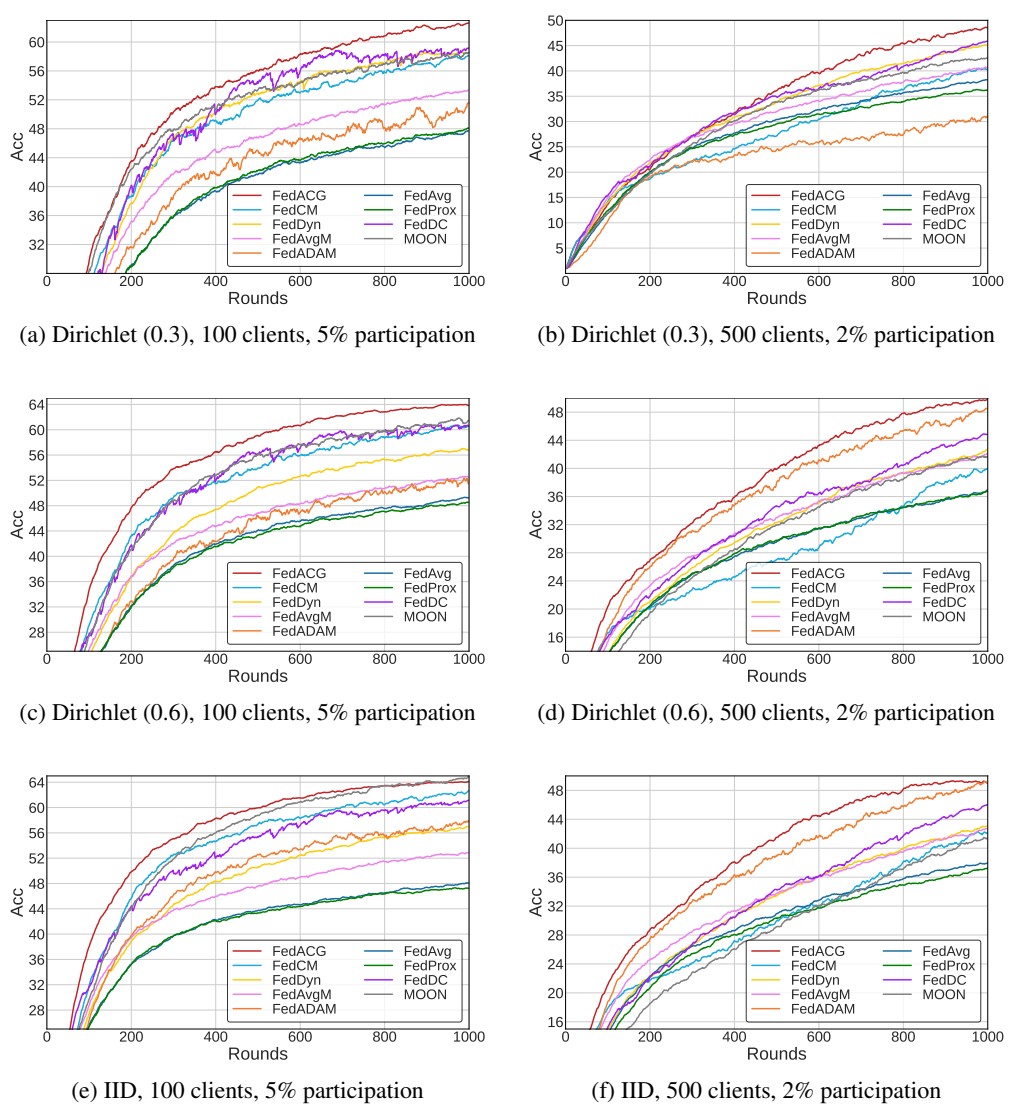

(a) Dirichlet (0.3), 100 clients, 5% participation

(b) Dirichlet (0.3), 500 clients, 2% participation

(c) Dirichlet (0.6), 100 clients, 5% participation

(d) Dirichlet (0.6), 500 clients, 2% participation

(e) IID, 100 clients, 5% participation

(f) IID, 500 clients, 2% participation

Figure 3: The convergence plots of FedACG and the baselines on CIFAR-100 with different federated learning scenarios.

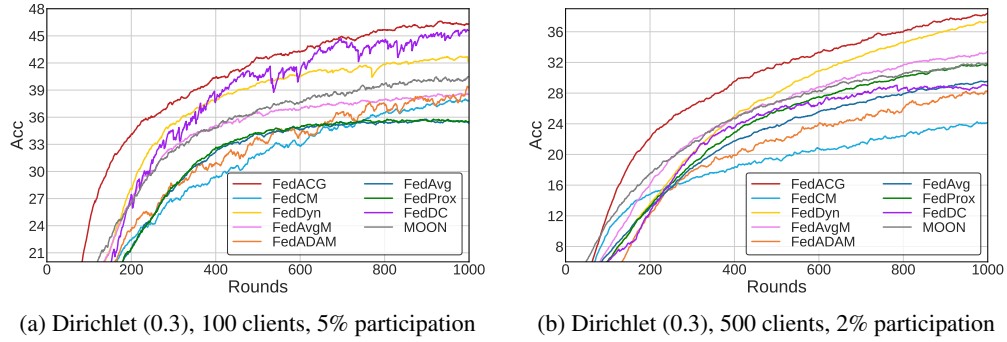

(a) Dirichlet (0.3), 100 clients, 5% participation

(b) Dirichlet (0.3), 500 clients, 2% participation

Figure 4: The convergence plots of FedACG and the baselines on Tiny-ImageNet with different federated learning scenarios.

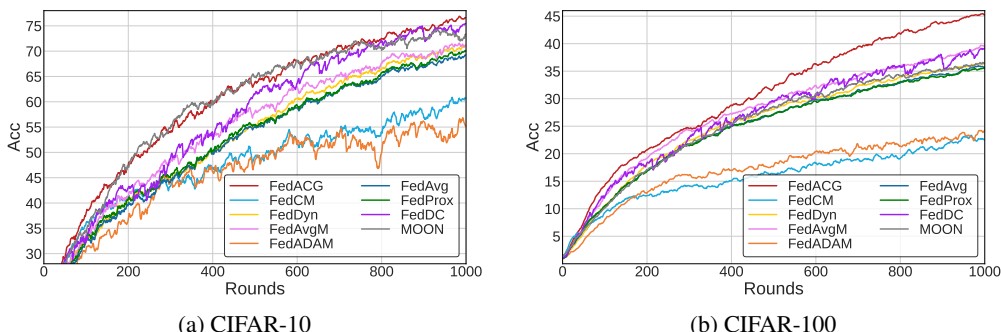

(a) CIFAR-10               (b) CIFAR-100

Figure 5: The convergence plots of FedACG and the baselines when participation rate is low (1%) for 500 clients on CIFAR-10 and CIFAR-100. The Dirichlet parameter is set to 0.3 for the experiments.

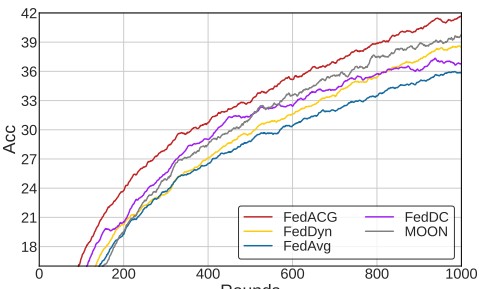

Figure 6: The convergence plots of FedACG, FedAvg, FedDyn, FedDC, and MOON on CIFAR-100 when the client set changes over dynamically: we sample 250 clients out of 500 clients as a candidate clients set at every 100 rounds over 10 stages on Dirichlet (0.3) split. 10 clients out of the sampled client set participate for the local training for each communication round. Dirichlet parameter is set to 0.3.

# E  DISCUSSION ABOUT THE DIFFERENCE BETWEEN FEDACG AND THE OTHER METHODS

**FedAvgM**   The unique characteristic of FedACG lies in client accelerated gradient. Although FedACG and FedAvgM use global momentum in common, FedACG broadcasts the accelerated global model by adding global momentum to the current global model ($\theta^{t-1} + \lambda m^{t-1}$) while FedAvgM (Hsu et al., 2020) broadcasts the current global model($\theta^{t-1}$) to each client as the initial point of the local model. To clarify the novelty of FedACG, we provide two pseudo-codes of FedACG and FedAvgM with local regularization term in Algorithm 2 and Algorithm 3, respectively. Figure 7 and Figure 8 also illustrates the server broadcasting and aggregation process of FedACG and FedAvgM, respectively.

**FedProx**   FedACG is a totally different method from FedProx for three reasons. First, FedACG utilizes the global momentum for server update as in Algorithm 2. Second, since FedACG uses client accelerated gradient, the local model's initial point is different from FedProx. From this, third, objective function of FedACG regularize the distance not between the local model and the previous global model (FedProx), but between the local model and the accelerated point.

# F  EFFECT OF LOCAL REGULARIZATION TERM

Table 9 shows the effect of local regularization term in FedAvg, FedAvgM, and FedACG. Note the role of the local regularization term is different in FedACG due to the acceleration term ($+\lambda m^{t-1}$) included in the message from the global model. We first observe that employing accelerated client gradient by adding global momentum to the current model plays a critical role for the performance gain. We also observe the effectiveness of the local regularization term in FedACG; Adding the local regularization term to the other baselines do not necessarily achieve performance gains in CIFAR-10 and CIFAR-100.

---

**Algorithm 2** FedACG

**Input:** $\beta$, $\lambda$, initial server model $\theta^0$, number of clients $N$, number of communication rounds $T$, number of local iterations $K$, local learning rate $\eta$

Initialize global momentum $m^0 = 0$
**for** each round $t = 1, 2, \ldots, T$ **do**
  Sample subset of clients $S_t \subseteq \{1, \ldots, N\}$
  Server sends $\theta^{t-1} + \lambda m^{t-1}$ for all clients $i \in S_t$
  **for** each client $i \in S_t$, **in parallel do**
    Initialize local model $\theta_{i,0}^t = \theta^{t-1} + \lambda m^{t-1}$
    **for** each local iteration $k = 1, 2, \ldots, K$ **do**
      Compute mini-batch loss $f_i(\theta_{i,k-1}^t) = \mathcal{L}_i(\theta_{i,k-1}^t) + \frac{\beta}{2}\|\theta_{i,k-1}^t - (\theta^{t-1} + \lambda m^{t-1})\|^2$
      $\theta_{i,k}^t = \theta_{i,k-1}^t - \eta \nabla f_i(\theta_{i,k-1}^t)$
    **end**
    $\Delta_i^t = \theta_{i,K}^t - (\theta^{t-1} + \lambda m^{t-1})$
    Client sends $\Delta_i^t$ back to the server
  **end**
  **In server:**
    $\Delta^t = \sum_{i \in S_t} \omega_i \Delta_i^t$
    $m^t = \lambda m^{t-1} + \Delta^t$
    $\theta^t = \theta^{t-1} + m^t$
**end**
**Return** $\theta^t$

---

---

**Algorithm 3** FedAvgM with local regularization

---

**Input:** $\beta$, $\lambda$, initial server model $\theta^0$, number of clients $N$, number of communication rounds $T$,
    number of local iterations $K$, local learning rate $\eta$
Initialize global momentum $m^0 = 0$
**for** each round $t = 1, 2, \ldots, T$ **do**
  Sample subset of clients $S_t \subseteq \{1, \ldots, N\}$
  Server sends $\theta^{t-1}$ for all clients $i \in S_t$
  **for** each client $i \in S_t$, **in parallel do**
   Initialize local model $\theta_{i,0}^t = \theta^{t-1}$
   **for** each local iteration $k = 1, 2, \ldots, K$ **do**
    Compute mini-batch loss $f_i(\theta_{i,k-1}^t) = \mathcal{L}_i(\theta_{i,k-1}^t) + \frac{\beta}{2}\|\theta_{i,k-1}^t - \theta^{t-1}\|^2$
    $\theta_{i,k}^t = \theta_{i,k-1}^t - \eta\nabla f_i(\theta_{i,k-1}^t)$
   **end**
   $\Delta_i^t = \theta_{i,K}^t - \theta^{t-1}$
   Client sends $\Delta_i^t$ back to the server
  **end**
  **In server:**
   $\Delta^t = \sum_{i \in S_t} \omega_i \Delta_i^t$
   $m^t = \lambda m^{t-1} + \Delta^t$
   $\theta^t = \theta^{t-1} + m^t$
**end**
**Return** $\theta^t$

---

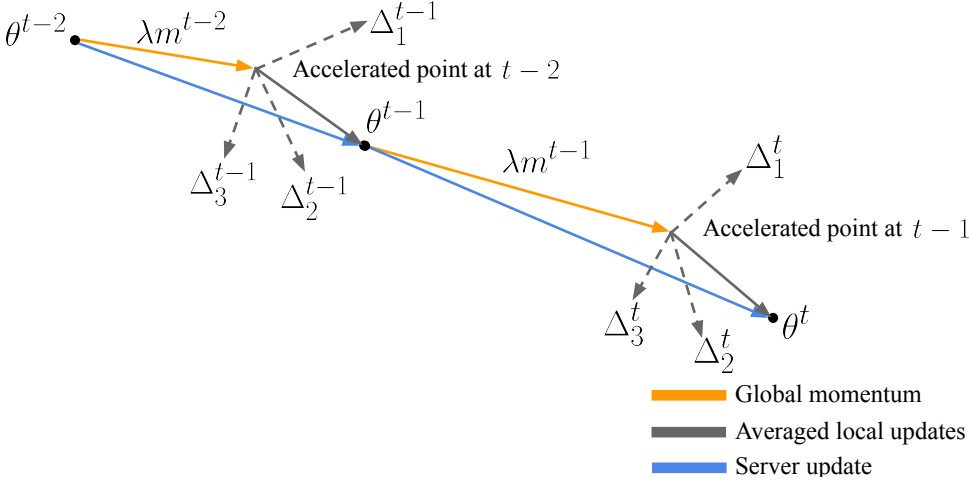

Figure 7: An illustration of the FedACG during two communication rounds

Table 9: Effect of local regularization term on different federtaed learning algorithms on CIFAR-10 and CIFAR-100 with 2% participation and 500 clients.

| Method | Server update w momentum | Accelerated client gradient | Local regularization | CIFAR-10 | CIFAR-100 |
|---|---|---|---|---|---|
| FedAvg | | | | 71.45 | 38.11 |
| FedProx | | | ✓ | 70.75 | 36.16 |
| FedAvgM | ✓ | | | 77.49 | 40.54 |
| FedAvgM w local reg | ✓ | | ✓ | 76.16 | 41.64 |
| FedACG w/o local reg | ✓ | ✓ | | 82.20 | 46.80 |
| FedACG | ✓ | ✓ | ✓ | **82.80** | **48.40** |

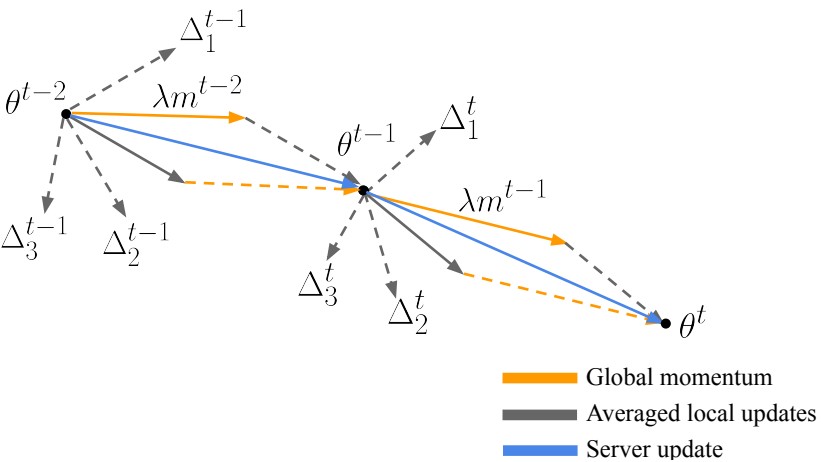

Figure 8: An illustration of the FedAvgM during two communication rounds

