# OpenReview forum: "Communication-Efficient Federated Learning with Accelerated Client Gradient"
_ICLR.cc/2023/Conference — Submitted to ICLR 2023_

### Official Review · Reviewer_fVhK · 2022-10-21

**Confidence:** 3
**Correctness:** 3
**Technical Novelty And Significance:** 2
**Empirical Novelty And Significance:** 3
**Recommendation:** 5

**Clarity, Quality, Novelty And Reproducibility:**

The paper is clearly written and easy to follow. However, it is lacking in novelty as FedACG only adds a regularization term on top of FedAvgM, and how this term works was not carefully discussed in the paper. Detailed hyperparameter setting together with the source code is provided and I believe it is easy to reproduce results in the paper.

**Strength And Weaknesses:**

Strengths:
- The proposed method is simple yet effective. Figure 1 gives a clear picture of the idea and makes the paper easy to follow.
- The experimental part is comprehensive. In detail, evaluation was conducted on three representative datasets under different data distribution. And it can be observed that FedACG performed consistently better among these settings. Ablation study to investigate individual components of the method is sufficient.
- A convergence analysis was provided to show that FedACG can guarantee the convergence of the model.

Weaknesses:

- The biggest concern of this paper is the difference between FedACG and FedAvgM [1]. It seems that FedACG merely adds a regularization term to reduce variance in local clients on top of FedAvgM, in which momentum was also leveraged to update the model in the server side. Then it is surprising to see that a regularizer can contribute to the performance gain significantly. More analysis and discussion of FedAvgM is necessary.
- To show FedACG achieves a better convergence rate theoretically, there should be a comparison among all other methods, i.e., the asymptotic convergence rate of methods such as FedAvg.
-
- It lacks a discussion about differential privacy of FedACG. It is important to a FL algorithm whether it can preserve differential privacy.

[1] Hsu, Tzu-Ming Harry, Hang Qi, and Matthew Brown. "Measuring the effects of non-identical data distribution for federated visual classification." arXiv preprint arXiv:1909.06335 (2019).

**Summary Of The Paper:**

The paper proposed a communication-efficient federated learning algorithm, FedACG to accelerate the convergence of training. Specifically, by adding a momentum to the aggregated weight and using this term as a regularizer, the authors show that the model can converge faster with higher accuracy theoretically and empirically.

**Summary Of The Review:**

Overall, the paper presented FedACG, a communication-efficient federated learning algorithm and demonstrated its advantages through extensive experiments. However, it was similar to a previous paper FedAvgM and no discussion about difference between two methods. Thus, it was marginally below the acceptance threshold.

---

> ### Author Response · Authors · 2022-11-09
> **Response to Reviewer fVhK**
>
> We thank you for your detailed and constructive feedback.
>
>
> **Q4.1\. Novelty of FedACG**
>
> A4.1\: Thank you for raising this issue. We want to emphasize that the main difference between FedACG and FedAvgM lies in what the server distributes, not in the regularization term. Let the $\beta$, which controls the degree of regularization, be zero. In this case, the local client’s initial local model is a distributed accelerated global model by the server, $\theta^{t-1} + \lambda m^{t-1}$. In FedAvgM, the local client’s initial local model is just the previous global model, $\theta^{t-1}$.
> Table A shows that the performance of FedACG without local regularization is 46.80% in CIFAR-100, while that of FedAvgM is 40.54%. And we also compare when both FedACG and FedAvgM have the local regularization term. We can observe that FedACG outperforms FedAvgM with local regularization by a large margin, demonstrating that the anticipatory update of FedACG makes a large difference in accuracy.
> For a more detailed discussion, please refer to our answers for Q1 by Reviewer Ym7U.
>
> | Method               | Server update w momentum | Accelerated client  gradient | Local regularization | CIFAR-10 | CIFAR-100 |
> | -------------------- | ------------------------ | ---------------------------- | -------------------- | -------- | --------- |
> | FedAvg               |                          |                              |                      | 71.45    | 38.11     |
> | FedProx              |                          |                              | ✓                    | 70.75    | 36.16     |
> | FedAvgM              | ✓                        |                              |                      | 77.49    | 40.54     |
> | FedAvgM w local reg  | ✓                        |                              | ✓                    | 76.16    | 44.64     |
> | FedACG w/o local reg | ✓                        | ✓                            |                      | 82.20    | 46.80     |
> | FedACG               | ✓                        | ✓                            | ✓                    | 82.80    | 48.40     |
>
>
>
>
> **Q4.2\. Discussion about differential privacy**
>
> A4.2\: From a privacy point of view, since FedACG does not require additional messages other than model parameters between the server and clients, it does not bring about any additional privacy concerns compared to FedAvg. Since it is the least communicated message that the general federated learning framework can have, we believe that FedACG can achieve the maximum level of differential privacy. (Please let us know if we misunderstood your question.)
>
>
>
> **Q4.3\. Convergence rate and its comparison with other methods**
>
> A4.3\: We will include the discussion about this issue during the rebuttal period.

---

> > ### Author Response · Authors · 2022-11-19
> > **Further Responses to Reviewer fVhK**
> >
> > **Q4.3. Convergence rate and its comparison with other methods.**
> >
> > We have proved FedACG convergence for non-convex functions taking into account the stochastic variance, and updated the details of the proof in our revision (Appendix A, Theorem 1).
> >
> > Here we present the theoretical convergence rate of FedACG for non-convex functions.
> >
> > **Theorem 1.** *(Convergence rate of FedACG) Suppose that local functions $F_{i}$ are non-convex and $L$-smooth.
> > Let $z^t = \theta^t + \frac{\lambda}{1 - \lambda} m^t$ for any $0 \leq \lambda < 1$.
> > Then, by setting $\eta = \text{min} (\frac{1-\lambda}{2 LK}, \frac{C}{\sqrt{T+1}})$, FedACG satisfies,*
> >
> >  $$ \text{min}_{t=0, \ldots, T} ~\mathbb{E} \| \| \nabla F(\theta^t + \lambda m^t)\| \|^2 \leq  \frac{2(F(z_0) )-F*)(1-\lambda)}{T+1} \text{max}(\frac{2LK}{1-\lambda},\frac{\sqrt{T+1}}{C})+\frac{C}{\sqrt{T+1}} B'',$$
> >
> > where
> > $$ B'' =\frac{1}{(1-\lambda)K}(
> >  ( (1+\frac{L^2K^4}{3})(1-\lambda) + \frac{ \lambda^4 L K^2}{2(1-\lambda)^2} + (1+ \frac{4N}{|S_t|(N-1)}(1-\frac{|S_t|}{N}))(LK^2 + \frac{L\lambda^4 K^2}{2(1-\lambda)^2})
> > )G^2 +  \frac{LK^2}{2} \big(2+\frac{\lambda^4}{(1-\lambda^2)} \big)\sigma^2.
> > )$$
> >
> >
> > The convergence rate of our algorithm on the non-convex setting is $\mathcal{O}(\frac{1}{\sqrt{T}})$, which matches the rate of FedAvg, FedADAM, and SCAFFOLD. Note that the convergence rate of FedAvg on the non-convex setting is offered by [Karimireddy et al., 2020]. Although FedDyn and FedDC are reported to have a better convergence rate in their proof, they do not consider the stochastic variance. Most of all, FedACG outperforms all the compared methods including FedDyn and FedDC in our experiments.
> >
> >
> > **Reference**
> >
> > [Karimireddy et al., 2020] Sai Praneeth Karimireddy et al. Scaffold: Stochastic controlled averaging for on-device federated learning. In ICML, 2020.

---

> > > ### Author Response · Authors · 2022-11-24
> > > **Convergence of FedACG for Strongly Convex and General Convex Functions**
> > >
> > > **Q4.3. Convergence rate and its comparison with other methods.**
> > >
> > > A4.3. As promised in our previous posting, we have proved the convergence of FedACG for convex and strongly convex functions, taking into account the stochastic variance,
> > >
> > > **Strongly convex:**
> > >
> > > **Theorem A.** *Suppose that local functions $F_{i}$ are strongly convex ($\mu > 0$) and $L$-smooth. Let $z^t = \theta^t + \frac{\lambda}{1 - \lambda} m^t$. For $\eta \leq  \frac{1-\lambda}{2LK}$ and $ 0 \leq \lambda <1$,*
> > >
> > > $$ \mathbb{E}[F(\bar{z}^T)] - F(z*) \leq \tilde{\mathcal{O}}(\frac  { (\frac{4N}{(N-1)|S_t|}(1-\frac{|S_{t}|}{N})+1)G^2 + \sigma^2}
> > > {\mu T} + \frac{L}{\mu^2 T^2}[(\frac{(1-\lambda)^2}{3}+\frac{\lambda^4}{(1-\lambda)^2}(\frac{4N}{(N-1)|S_t|}(1-\frac{|S_{t}|}{N})+1))G^2 +(\frac{(1-\lambda)^2}{3}+\frac{\lambda^4}{(1-\lambda)^2})\sigma^2]+\mu \mathbb{E}||z^{0}-z*||^2 exp(- \frac{\mu}{L}T) ) $$
> > >
> > >
> > > **General convex:**
> > >
> > > **Theorem B.** *Suppose that local functions $F_{i}$ are general convex ($\mu = 0$) and $L$-smooth. Let $z^t = \theta^t + \frac{\lambda}{1 - \lambda} m^t$. For $\eta \leq  \frac{1-\lambda}{2LK}$ and $ 0 \leq \lambda <1$,*
> > >
> > >
> > > $$ \mathbb{E}[F(\bar{z}^T)] - F(z*) \leq \mathcal{O}(\sqrt{\frac{((\frac{4N}{(N-1)|S_t|}(1-\frac{|S_{t}|}{N})+1)G^2 +\sigma^2)\mathbb{E}||z^{0}-z*||^2}{T}} + \sqrt[3]{\frac{L \mathbb{E}||z^{0}-z*||^2}{ T^2}[(\frac{(1-\lambda)^2}{3}+\frac{\lambda^4}{(1-\lambda)^2}(\frac{4N}{(N-1)|S_t|}(1-\frac{|S_{t}|}{N})+1))G^2 +(\frac{(1-\lambda)^2}{3}+\frac{\lambda^4}{(1-\lambda)^2})\sigma^2]} + \frac{L}{T} \mathbb{E}||z^{0}-z*||^2)$$
> > >
> > > We proved them using the lemmas used for non-convex case (Appendix A.1 in the revision), and lemma 1 and lemma 2 in SCAFFOLD [Karimireddy et al., 2020]. We will include the details of the proofs in the final version.
> > >
> > > Our new proofs consider stochastic variance with strongly convex and convex functions, and show that FedACG converges $\tilde{\mathcal{O}}(\frac{1}{T})$ and $\mathcal{O}(\frac{1}{\sqrt{T}})$ for strongly convex and convex functions, respectively, which matches the rates of SCAFFOLD. As with the non-convex case, although FedDyn and FedDC are reported to have better convergence rates in their proof, they do not consider the stochastic variance. Please refer to our response **A1.2** to reviewer KHhq for more discussion on the convergence of FedACG.

---

> > ### Comment · Reviewer_fVhK · 2022-12-05
> > **Post-rebuttal comments**
> >
> > Thanks for the clarification of the proposed method and FedAvgM. Based on my understanding, FedACG anticipated the parameter at the next step with the current momentum and sent it to local clients. However, I shared the same concern with other reviewers that such a modification can improve the performance significantly. A more rigorous analysis is necessary.

---

> > > ### Author Response · Authors · 2022-12-05
> > > **Why FedACG outperforms FedAvgM**
> > >
> > > Thank you for your comment.
> > >
> > > It is generally good to use global momentum, which is the exponential sum of previous global gradients, for the server update. However, the stability of the heterogeneous federated learning algorithm depends on where and how momentum is used, and our learning strategy is simple yet more effective to handle this issue than FedAvgM. Since only a subset of the clients participates in the training for each round and the norm of local updates may be large due to K local gradient steps, high data heterogeneity in the federated learning can cause the global momentum to be noisy. Unlike FedAvgM, where the only server knows the global momentum, in FedACG, the clients, the actual learners, start their local training at the anticipated point $\theta^{t-1}+ \lambda m^{t-1}$ and can make up for the previous global momentum $\lambda m^{t-1}$ through their local training. This difference in the initial point seems to allow FedACG to change $m^t$ more responsive way, letting it behave more stable than FedAvgM in many federated learning settings. The effect of the difference for each communication round may be accumulated as the communication progresses. We believe that our comprehensive experiments in various federated settings and ablation studies of FedACG sufficiently validate the effectiveness of the FedACG compared to FedAvgM.

---

> > > > ### Author Response · Authors · 2022-12-09
> > > > **Further analysis between FedACG and FedAvgM**
> > > >
> > > > Thank you for your constructive comments.
> > > >
> > > >
> > > > To further validate the effectiveness of the different local initial point in FedACG, we plot the accuracy of FedACG and FedAvgM on CIFAR-10 in the moderate-scale setting without smoothing (exponential moving average), and we provide the result in this [link](https://anonymous.4open.science/r/paper1852_clarification-B831/cifar10_moderate.png). We observe that FedACG not only outperforms FedAvgM, but also converges with a smaller level of oscillation than FedAvgM throughout the training procedure. Please refer to our [response to reviewer KHhq](https://openreview.net/forum?id=de-_FHXQ4--&noteId=puFMstGQswZ) for the detailed discussion.
> > > >
> > > >
> > > > We appreciate your time and efforts in reviewing all these comments and the revision.
> > > > We have provided detailed responses to address your concerns and questions. If our response has addressed all your concerns, we hope you will reflect this in your final review and score. We would like to provide additional discussions during the remaining discussion period if you have further questions.

---

### Official Review · Reviewer_Ym7U · 2022-10-25

**Confidence:** 3
**Correctness:** 3
**Technical Novelty And Significance:** 2
**Empirical Novelty And Significance:** 2
**Recommendation:** 6

**Clarity, Quality, Novelty And Reproducibility:**

Clarity and quality can be improved as it pertains to the empirical evaluation concerns mentioned. The proposed technique FedACG is related to FedAvgM [Hsu et al. 2019] as it pertains to using a momentum term, which I do not see mentioned and discussed in Section 2 Related Work. My understanding is that FedACG and FedAvgM computes the momentum at the sever-side. The difference being FedAvgM incorporates the term to the global parameters, whereas FedACG utilizes it as a regularization term on the client-side. However, it is used and compared to in the empirical evaluation. The results are strikingly different, and some analysis would be helpful.

**Strength And Weaknesses:**

The proposed technique is simple and the idea is intuitive. However, my main concern is with some of the empirical evaluation. Specifically:

1. How many trials were used in running the experiments?
2. Are the results in Table 1 (a) and (b) under the non-IID setting since the description in the table says "The Dirichlet parameter is commonly set to 0.3"?
3. The way the Dirichlet distribution sampling is used in the paper seems different from [Hsu et al. 2019], where the data is drawn from Dirchlet($\alpha,\boldsymbol{p}$), where $\boldsymbol{p}$ characterizes a prior class distribution over $N$ classes, and $\alpha{>}0$ is a concentration parameter controlling the identicalness among classes.
4. Do all the accuracy numbers use the exponential moving average with 0.9 coefficient (meaning older observations are discounted faster)?
5. In Table 1 (a): for MOON (CIFAR-100), 500R shows Acc. of 83.32%, but 81% is achieved at 543 rounds? In Table 2: for FedDC (CIFAR-10), 500R shows Acc. of 60.56%, but 64% is achieved at 509 rounds. This means over 3% was gained in 9 rounds?
6. Table 6 shows that accuracy is sensitive to the shown different settings to $\lambda$, but stable to the show different settings to $\beta$. This is surprising, and somewhat counter intuitive from the perspective of $\ell_2$-regularization.
7. Table 6 shows the more realistic FL setting with feature skew and data imbalance. The results for FedACG compared to FedDC, FedAvg, FedAvgM are close enough such that it is not clear on the statistical significance of the result. I would be curious to see incorporating data imbalance in the main experimental results.
8. The empirical evaluations are limited to image datasets and one neural network model.



**Summary Of The Paper:**

The paper (FedACG) proposes the following additive regularization term to the learning objective function for federated learning clients:

$$
\quad\quad \frac{\beta}{2}||\theta - (\theta^{t-1} + \lambda m^{t-1}) ||^2
$$

where $\beta$ and $\lambda$ are tuning parameters. $\theta$ is the model parameters. The $m$ momentum term is calculated at the server, which is the difference between the sum of weighted parameter values received  from clients at round $t$ and round $t{-}1$. This is similar to $\ell_2$-regularization, but with a gradient acceleration (or momentum term).

The paper uses ResNet-18 and evaluates on datasets: CIFAR-10, CIFAR-100, Tiny-ImageNet, FEMNIST, and CelebA. IID data is generated via random sampling without replacement. Non-IID data is generated via sampling label ratios from a Dirichlet distribution. All clients hold the same number of training data.

Communication efficiency is captured by (reduced) number of communication rounds to achieve a particular level of test accuracy.


**Summary Of The Review:**

The proposed technique is simple and appears effective given the empirical results. However, the empirical evaluation can be improved.

Update: I thank the authors for answering my questions. I will maintain my review. However, having read through the other reviews (and replies) I think that the comparison with (at least) FedAvgM needs to be further clarified and highlighted.

---

> ### Author Response · Authors · 2022-11-09
> **Response to Reviewer Ym7U(1/2)**
>
> We truly appreciate your positive and constructive comments, and we will reflect on your feedback thoroughly. Here are the responses to the issues.
>
> **Q3.1\. Discussion about FedAvgM**
>
> A3.1\: Our main contribution is to add a momentum factor $m^t$ to the current server model and each participating client optimizes its local loss from the accelerated point, while FedAvgM naively switches server SGD into SGD with momentum (refer to Section 4.2 in [Hsu et al., 2019] and Section 5.1 in [Hsu et al., 2020]). For better understanding, we have provided pseudo codes and simple illustrations of both methods in our revision (refer to Algorithms 2 and 3, and Figures 7 and 8 in Appendix E in our revision). While both methods use server momentum in the server-side aggregation, our contribution is still significant because taking lookahead through the global momentum for the local updates is sufficiently unique and leads to outstanding performance without increasing communication costs at all.
> Table A presents that the proposed accelerated client gradient has a more critical impact on accuracy than other components, i.e., FedACG without the local regularization term in Eq (2) in the main paper shows 4.71% and 6.26% accuracy gain than FedAvgM on CIFAR-10 and CIFAR-100, respectively.
>
>
> ***Table A.*** Contribution of individual components in FedACG at 1K rounds on CIFAR-10 and CIFAR-100 with 2% participation and 500 clients
>
>
>
> | Method               | Server update w momentum | Accelerated client  gradient | Local regularization | CIFAR-10 | CIFAR-100 |
> | -------------------- | ------------------------ | ---------------------------- | -------------------- | -------- | --------- |
> | FedAvg               |                          |                              |                      | 71.45    | 38.11     |
> | FedProx              |                          |                              | ✓                    | 70.75    | 36.16     |
> | FedAvgM              | ✓                        |                              |                      | 77.49    | 40.54     |
> | FedAvgM w local reg  | ✓                        |                              | ✓                    | 76.16    | 44.64     |
> | FedACG w/o local reg | ✓                        | ✓                            |                      | 82.20    | 46.80     |
> | FedACG               | ✓                        | ✓                            | ✓                    | 82.80    | 48.40     |
>
>
>
> **Q3.2\. Sensitivity of FedACG on $\beta$**
>
> A3.2\: Since our main contribution is to employ the accelerated gradient using momentum, it is natural that FedACG achieves high performance when $\lambda$ is close to 1. Table 5(a) of the main paper shows the impact of $\lambda$ and we can notice that the overall performance is pretty stable when $\lambda \geq 0.75$. The performance of our algorithm looks stable with respect to the variation of $\beta$, but this is because the global momentum (gradient acceleration) has already been applied to the model and $\beta$ makes a minor contribution to performance gains.
>
>
> **Q3.3\. Evaluation in the presence of data imbalance:**
>
> A3.3\: We will include the experiments that incorporate data imbalance.
>
> **Q3.4\.Need evaluation on other tasks, and other backbone networks:**
>
> A3.4\: We will include the evaluation on different tasks with different backbone architectures.
>
> **Q3.5\. Constructing non-iid data with Dirichlet distribution:**
>
> A3.5\: We followed the conventional strategy of existing works for constructing non-iid data with Dirichlet distribution. Specifically, we apply latent Dirichlet allocation over labels of the dataset. Each client has an individual multinomial distribution drawn from a symmetric Dirichlet distribution with parameters $\alpha$ and $p$, and its training examples are sampled from the multinomial distribution. Therefore, each client has non-iid data even if $\alpha$ is set to the common value. Since $p$ is a prior distribution decided by datasets, the scale of $\alpha$ controls the identicalness among clients. With $\alpha \rightarrow \infty$, all clients have iid data distribution; with $\alpha \rightarrow 0$, each client holds examples from only one class chosen at random.
>
>
> **References**
>
> [Hsu et al., 2019] TMH. Hsu et al., Measuring the effects of non-identical data distribution for federated visual classification. arXiv preprint arXiv:1909.06335, 2019.
>
> [Hsu et al., 2020]  TMH. Hsu et al., Federated Visual Classification with Real-World Data Distribution, ECCV, 2020

---

> > ### Author Response · Authors · 2022-11-09
> > **Response to Reviewer Ym7U(2/2)**
> >
> > **Q3.6\. Do all the accuracy numbers use the exponential moving average with 0.9 coefficient (meaning older observations are discounted faster)?:**
> >
> > A3.6\:   We use the following update rule: $\text{Acc}\_{\text{ema}}= 0.1 * \text{Acc}\_{\text{current}} + 0.9 * \text{Acc}_{\text{ema}}$
> >
> > **Q3.7\. Typos in Tables 1 and 2**
> >
> > A3.7\: In Table 1 (a): for results of a few methods (FedDyn, MOON, FedDC, and FedACG), the rows for the rounds to 81% were miscalculated. And, in Table 2: for FedDC and MOON (CIFAR-10), the required rounds to 64% and 68% should be (610, 681) and (491, 645), respectively. We apologize for these typos and would like to emphasize that all the other numbers in the paper are correct and the typos do not damage our argument in Section 4 of the main paper. We have uploaded the revision.
> >
> > **Q3.8\. How many trials were used in running the experiments?**
> >
> > A3.8\: The results are reported from models trained from the same initialization (random seed = 1). We will include the results from the average of multiple seeds.

---

> > > ### Author Response · Authors · 2022-11-19
> > > **Further Responses to Reviewer Ym7U**
> > >
> > > **Q3.4. Evaluation on different tasks and with different backbone networks**
> > >
> > > A3.4: We evaluate FedACG for a different task suggested by Reviewer Ym7U, language modeling on ShakeSpeare, and present the results in Table B below. In this experiment, we use a small-sized dataset from LEAF and use LSTM as a backbone network. The participation rate for each round is set to 5%. We can observe that the proposed method is also effective on other tasks (language modeling) besides image classification. Note that, the model was not trained well in FedCM despite tuning the hyperparameters.
> > >
> > > We want to report back on our responses to experiments on different backbone networks. We use different backbone networks according to our benchmarks: simple CNN on {FMNIST and CelebA}, ResNet-18 on {CIFAR-10, CIFAR-100, and Tiny-ImageNet}, and LSTM for language modeling. The proposed algorithm outperforms the compared method in most settings regardless of the backbone networks.
> > >
> > > ***Table B.***  Results of FedACG and the compared methods on ShakeSpeare.
> > >
> > > | Method  |  ACC  |       | Rounds |       |
> > > |---------|:-----:|:-----:|:------:|:-----:|
> > > |         |  500R | 1000R |   42%  |  45%  |
> > > | FedAvg  | 45.01 | 46.55 |   94   |  500  |
> > > | FedProx | 45.09 | 46.29 |   99   |  477  |
> > > | FedAvgM | 44.63 | 45.91 |   63   |  690  |
> > > | FedADAM | 44.89 |  44.3 |   68   | 1000+ |
> > > | FedDyn  | 39.23 |  44.1 |   749  | 1000+ |
> > > | MOON    | 42.02 | 42.65 |   499  | 1000+ |
> > > | FedCM   |   -   |   -   |    -   |   -   |
> > > | FedDC   | 30.62 | 44.27 |   926  | 1000+ |
> > > | FedACG  | 46.36 | 48.23 |   57   |  290  |
> > >
> > >
> > > **Q3.3. Experiments in the presence of data imbalance**
> > >
> > > A3.3:  Since the ShakeSpeare benchmark includes a large data imbalance between clients [McMahan et al., 2017], the results in Table A also demonstrate the effectiveness of FedACG over the imbalanced data. We believe that the reason why Table 6 in the main paper shows similar performance between algorithms is not because of the presence of the data imbalance, but because the LEAF benchmark tackles the easy tasks, e.g., binary classification on CelebA and 10-class classification on MNIST, so all algorithms converge sufficiently despite data heterogeneity. Nevertheless, Table 6 in the main paper shows that FedACG outperforms the compared algorithms, which still validates the effectiveness of FedACG on federated settings in the presence of data imbalance and feature skewness in terms of accuracy and convergence speed.
> > >
> > > We do hope that we were able to address all questions and concerns in our detailed responses. Otherwise, please let us know about the remaining concerns and where you think we could further improve. Such post-response feedback would be greatly appreciated.
> > >
> > > **Reference**
> > >
> > > [McMahan et al., 2017] Brendan McMahan et al., Communication-efficient learning of deep networks from decentralized data. In AISTATS, 2017.

---

> > > > ### Author Response · Authors · 2022-11-24
> > > > **Further Responses to Reviewer Ym7U**
> > > >
> > > >
> > > > ***Table D.*** Effect of low participation rate, 1% over 500 clients, for FedACG and the compared methods on CIFAR-10 and CIFAR-100.
> > > > | Method  | CIFAR10 |       |      |      | CIFAR100 |       |      |      |
> > > > | ------- | :--------: | :-----: | :----: | :----: | :---------: | :-----: | :----: | :----: |
> > > > |         | 500R     | 1000R | 64%  | 68%  | 500R      | 1000R | 30%  | 35%  |
> > > > | FedAvg  | 54.47    | 68.73 | 786  | 953  | 26.41     | 34.67 | 684  | 983  |
> > > > | FedProx | 54.89    | 68.96 | 787  | 935  | 25.98     | 33.80 | 730  | 988  |
> > > > | FedAvgM | 57.93    | 71.37 | 669  | 820  | 27.14     | 36.59 | 654  | 895  |
> > > > | FedADAM | 48.02    | 55.07 | 1000+ | 1000+ | 17.43     | 23.60 | 1000+ | 1000+ |
> > > > | FedDyn  | 56.73    | 72.06 | 664  | 790  | 27.10     | 35.53 | 636  | 925  |
> > > > | MOON    | **64.12**    | 73.38 | 508  | 652  | 27.48     | 36.10 | 613  | 924  |
> > > > | FedCM   | 50.17    | 60.37 | 1000+ | 1000+ | 15.95     | 22.60 | 1000+ | 1000+ |
> > > > | FedDC   | 60.98    | 75.29 | 590  | 679  | 29.71     | 38.70 | 506  | 745  |
> > > > | FedACG  | 63.68    | **76.52** | **502**  | **628**  | **31.23** | **45.41** | **471**  | **592**  |

---

> > > > > ### Author Response · Authors · 2022-12-09
> > > > > **Look forward to hearing your feedback**
> > > > >
> > > > > We appreciate your time and efforts in reviewing our paper, and we have revised the manuscripts and provided detailed responses to address your concerns. As the discussion period is only a few days left, we hope you respond to our rebuttal if you have any remaining concerns or questions, or if all your concerns have been resolved well. We would be happy to continue the discussion during the remaining discussion period.

---

> > > > ### Author Response · Authors · 2022-11-24
> > > > **Further Responses to Reviewer Ym7U**
> > > >
> > > > **Q3.8. How many trials were used in running the experiments?**
> > > >
> > > > A3.8: As promised in our previous posting, we provide results of the main experiments averaged by running three different seeds to analyze the true effectiveness of our framework in Table B to D. The results present that the proposed algorithm outperforms the compared federated learning algorithms with the meaningful margins in terms of accuracy and convergence speed for most cases, except that, compared with the results of Table 1 (a) in the main paper, FedDC outperforms FedACG in rounds reaching 85% on CIFAR10 in the moderate-scale setting. Note that, FedDC requires $2 \times$ communication costs for each communication round respectively since it communicates the current model and the associated gradient information per round, while FedACG only requires the model parameters.
> > > >
> > > > ***Table B.*** Results on the moderate-scale experiments where the number of clients and participation rate are set to 100 and 5%, respectively.
> > > > | Method  | CIFAR10 |       |     |      | CIFAR100 |       |     |      | Tiny-ImageNet |       |     |      |
> > > > | ------- | :--------: | :-----: | :---: | :----: | :---------: | :-----: | :---: | :----: | :-------------: | :-----: | :---: | :----: |
> > > > |         | 500R     | 1000R | 81% | 85%  | 500R      | 1000R | 47% | 55%  | 500R          | 1000R | 35% | 38%  |
> > > > | FedAvg  | 75.91    | 83.27 | 748 | 1000+ | 42.17     | 47.79 | 901 | 1000+ | 33.32         | 34.99 | 882 | 1000+ |
> > > > | FedProx | 75.45    | 83.41 | 735 | 1000+ | 42.60     | 48.32 | 865 | 1000+ | 33.47         | 34.80 | 871 | 1000+ |
> > > > | FedAvgM | 81.35    | 86.19 | 493 | 809  | 46.09     | 52.37 | 566 | 1000+ | 35.83         | 37.16 | 509 | 997  |
> > > > | FedADAM | 73.62    | 82.33 | 898 | 1000+ | 43.97     | 51.07 | 729 | 1000+ | 33.08         | 38.75 | 648 | 910  |
> > > > | FedDyn  | 85.58    | 88.35 | 367 | 530  | 48.63     | 55.92 | 435 | 884  | 37.13         | 40.92 | 362 | 577  |
> > > > | MOON    | 83.17    | 86.16 | 381 | 687  | 52.43     | 58.68 | 333 | 645  | 33.97         | 38.58 | 389 | 556  |
> > > > | FedCM   | 79.57    | 84.13 | 615 | 1000+ | 51.76     | 58.31 | 321 | 746  | 32.22         | 38.06 | 695 | 982  |
> > > > | FedDC   | **86.55**    | 87.92 | 325 | **461**  | 54.86     | 59.82 | 301 | 517  | 40.86         | 45.55 | 319 | 383  |
> > > > | FedACG  | 85.12    | **89.30** | **317** | 476  | **55.82**     | **61.68** | **268** | **453**  | **42.49**  | **46.67** | **229** | **310**  |
> > > >
> > > >
> > > > ***Table C.*** Results on the large-scale experiments where the number of clients and participation rate are set to 500 and 2%, respectively.
> > > > | Method  | CIFAR10 |       |      |      | CIFAR100 |       |      |      | Tiny-ImageNet |       |        |        |
> > > > | ------- | :--------: | :-----: | :----: | :----: | :---------: | :-----: | :----: | :----: | :-------------: | :-----: | :------: | :------: |
> > > > |         | 500R     | 1000R | 73%  | 77%  | 500R      | 1000R | 36%  | 40%  | 500R          | 1000R | 24% | 30% |
> > > > | FedAvg  | 59.30    | 71.94 | 979  | 1000+ | 29.40     | 37.12 | 908  | 1000+ | 23.89         | 29.57 | 505    | 982    |
> > > > | FedProx | 58.32    | 71.02 | 1000 | 1000+ | 28.65     | 35.52 | 989  | 1000+ | 24.68         | 30.51 | 473    | 905    |
> > > > | FedAvgM | 65.82    | 77.26 | 755  | 945  | 31.51     | 40.21 | 733  | 978  | 26.85         | 33.15 | 413    | 693    |
> > > > | FedADAM | 61.69    | 70.49 | 1000+ | 1000+ | 24.53     | 33.03 | 1000+ | 1000+ | 22.02         | 27.56 | 667    | 1000+   |
> > > > | FedDyn  | 66.24    | 79.54 | 681  | 855  | 30.13     | 39.41 | 795  | 976  | 24.73         | 29.51 | 461    | 1000+   |
> > > > | MOON    | 69.66    | 78.38 | 600  | 833  | 32.23     | 41.63 | 658  | 885  | 27.10         | 32.52 | 380    | 687    |
> > > > | FedCM   | 69.01    | 76.53 | 749  | 1000+ | 27.82     | 39.90 | 825  | 976  | 17.36         | 21.80 | 992    | 1000+   |
> > > > | FedDC   | 73.16    | **84.07** | 493  | 645  | 35.29     | 46.96 | 536  | 691  | 24.36         | 27.40 | 506    | 1000+   |
> > > > | FedACG  | **74.67**    | 83.05 | **460**  | **571**  | **35.53**     | **47.47** | **522**  | **649**  | **29.95**  | **37.24** | **276**    | **507**    |

---

> ### Author Response · Authors · 2022-12-12
> **Response to the post-rebuttal update**
>
> We appreciate your time to read all our revision, responses, and extensive discussions with other reviewers. While we provided detailed comparisons between FedACG and FedAvgM with additional illustrations (Figure 7 and Figure 8, and Algorithm 2 and 3 in the revision), as suggested by reviewer Ym7U, we further provided [detailed explanations](https://openreview.net/forum?id=de-_FHXQ4--&noteId=GwlfIvqJH9q) why the proposed framework is more suitable for heterogeneous federated learning than FedAvgM and [empirical analysis](https://openreview.net/forum?id=de-_FHXQ4--&noteId=puFMstGQswZ) that support our claim. We will reflect on the suggestions and our responses in the final copy. Thanks for your careful feedback and feel free to let us know if you have any other questions.

---

### Official Review · Reviewer_bEuE · 2022-10-25

**Confidence:** 4
**Clarity, Quality, Novelty And Reproducibility:** Look good to me.
**Correctness:** 3
**Technical Novelty And Significance:** 2
**Empirical Novelty And Significance:** 3
**Recommendation:** 5

**Strength And Weaknesses:**

Strength:
1. Propose practical algorithm that considers more realistic and strict FL settings
2. Outstanding performance in terms of communication efficiency and model performance over many baselines
3. The paper is well-written and easy to understand.

Weakness
1. The proposed algorithm FedACG is very similar to FedProx (Li et al., 2020) in the sense that FedACG has the additional hyperparameter \lambda. For example, when \lambda = 1, FedACG becomes FedProx. It would be great if this paper could elaborate on the relationship between FedACG and FedProx, and why introducing this additional hyperparameter \lambda is beneficial.
2.  Following on from the first point, introducing this additional hyperparameter \lambda results in better performance but it’s not without any cost. Typically, we want to keep the number of hyperparameters as minimal as possible, because hyperparameter tuning is very costly. Actually, according to Table 6(a), the performance of FedACG varies a lot depending on the value of \lambda.
3. For the convergence analysis (i.e., Theorem 1), I am not sure why we need to assume 0.5 < \lambda < 1, which seems to be a very strict assumption for \lambda because it’s practically possible for 0 < \lambda <= 1 (or even \lambda > 1 if we want more emphasis on the momentum m). Is it possible to derive a similar convergence rate when 0< \lambda < 0.5? In addition, when \lambda = 1,  FedACG becomes FedProx. It would be great if the convergence analysis can recover the analysis of FedProx when  \lambda = 1.
4. Lastly, [1] (see reference list below) has also considered using momentum for FL, and may need to be discussed in this paper.

Reference.
[1] Das, Rudrajit, et al. "Faster non-convex federated learning via global and local momentum." Uncertainty in Artificial Intelligence. PMLR, 2022.


**Summary Of The Paper:**

This paper presents a communication-efficient federated optimization algorithm that deals with client heterogeneity under large-scale and low-participation FL settings. The proposed algorithm, FedACG, leverages global momentum and local regularization. This paper demonstrates outstanding performance in terms of communication efficiency and robustness to client heterogeneity.

**Summary Of The Review:**

The paper proposed an interesting method for FL. But I am afraid that it has several major weaknesses to be addressed.

---

> ### Author Response · Authors · 2022-11-09
> **Response to Reviewer bEuE**
>
> We truly appreciate your constructive comments, and we will reflect on your feedback thoroughly. Here are the responses to the issues.
>
>
> **Q2.1\. Misunderstanding about FedACG**
>
> A2.1\: Our main contribution is to make the server broadcast the global model integrated with a global momentum factor $m^t$, where $\lambda$ determines how much the momentum is integrated. Since the client-side optimization of our method (FedACG) is similar to FedProx, our algorithm becomes identical to FedProx when $\lambda$ is equal to 0, not 1. Although a special case of FedACG is equivalent to FedProx, our contribution is still significant because accelerating each client’s initialization point with a global momentum using a non-zero $\lambda$ is sufficiently unique and leads to outstanding performance without increasing communication costs at all. We again emphasize that FedACG is the same as FedProx when $\lambda = 0$, which has a completely different meaning from the case that the two algorithms become identical when $\lambda \neq 0$. The comment by Reviewer bEuE about the similarity with FedProx is probably because he/she misunderstood the server’s role in FedACG.
>
>
> **Q2.2\. Cost for tuning hyperparameters**
>
> A2.2\: The *cost* in the statement that “FedACG do not have additional cost” is the *communication cost per each communication round*, and it is true that FedACG, like FedAvg, only communicates model parameters between server and client, so it does not increase communication cost compared to FedAvg.
>
> We agree that there may be a cost for tuning the newly introduced hyperparameters ($\lambda$ and $\beta$). However, $\lambda$ may not require exhaustive tuning since the range of $\lambda$ values does not deviate from the coefficient in general momentum update, while FedACG shows quite consistent performance over a wide range of $\beta$ values. The benefit outweighs its cost since adding $\lambda$ results in a large performance gain in various federated settings.
>
> **Q2.3\. Discussion about FedGLOMO**
>
> A2.3\: We agree FedGLOMO [Rudrajit et al., 2022] is related to our work in the sense of using global momentum for federated learning. However, unlike FedGLOMO which adopts the variance reduction technique (STORM) in both client and server updates, we adopt an accelerated gradients technique at the client side with the global momentum, which enables each client to look ahead to the trajectory of global gradients. In addition, FedGLOMO requires twice the communication cost because both the momentum term and the model parameter need to be communicated. We will add the discussion in our revised paper.
>
> **Reference**
>
> [Rudrajit et al., 2022] Das Rudrajit et al. "Faster non-convex federated learning via global and local momentum." Uncertainty in Artificial Intelligence. PMLR, 2022.

---

> > ### Author Response · Authors · 2022-11-19
> > **Further Responses to Reviewer bEuE**
> >
> >
> > **Q2.4. Constraints of $\lambda$ for the convergence of FedACG**
> >
> > As suggested by Reviewer KHhq, We have proved the convergence of FedACG for the non-convex case, taking into account the stochastic variance, and updated the details of the proof in our revision (Appendix A, Theorem 1).
> >
> > In our theoretical analysis of FedACG in a non-convex function, FedACG converges under $0 \leq \lambda < 1$, which is the same with the constraint for the coefficient of general SGD with momentum. Note that When $\lambda > 1$, our algorithm diverges because, intuitively, the momentum $m^t$ over which the gradient accumulates grows exponentially by $\lambda$. The convergence analysis on the convex case with considering the stochastic variance is similar to that of the non-convex case (following Lemma 1 in the Appendix), so we will also include the discussion about the convex case during the discussion period.

---

> > > ### Author Response · Authors · 2022-11-24
> > > **Convergence of FedACG for Strongly Convex and General Convex Functions**
> > >
> > > **Q2.4. Constraints of $\lambda$ for the convergence of FedACG**
> > >
> > > A2.4. As promised in our previous posting, we have proved the convergence of FedACG for convex and strongly convex functions, taking into account the stochastic variance,
> > >
> > > **Strongly convex:**
> > >
> > > **Theorem A.** *Suppose that local functions $F_{i}$ are strongly convex ($\mu > 0$) and $L$-smooth. Let $z^t = \theta^t + \frac{\lambda}{1 - \lambda} m^t$. For $\eta \leq  \frac{1-\lambda}{2LK}$ and $ 0 \leq \lambda <1$,*
> > >
> > > $$ \mathbb{E}[F(\bar{z}^T)] - F(z*) \leq \tilde{\mathcal{O}}(\frac  { (\frac{4N}{(N-1)|S_t|}(1-\frac{|S_{t}|}{N})+1)G^2 + \sigma^2}
> > > {\mu T} + \frac{L}{\mu^2 T^2}[(\frac{(1-\lambda)^2}{3}+\frac{\lambda^4}{(1-\lambda)^2}(\frac{4N}{(N-1)|S_t|}(1-\frac{|S_{t}|}{N})+1))G^2 +(\frac{(1-\lambda)^2}{3}+\frac{\lambda^4}{(1-\lambda)^2})\sigma^2]+\mu \mathbb{E}||z^{0}-z*||^2 exp(- \frac{\mu}{L}T) ) $$
> > >
> > >
> > > **General convex:**
> > >
> > > **Theorem B.** *Suppose that local functions $F_{i}$ are general convex ($\mu = 0$) and $L$-smooth. Let $z^t = \theta^t + \frac{\lambda}{1 - \lambda} m^t$. For $\eta \leq  \frac{1-\lambda}{2LK}$ and $ 0 \leq \lambda <1$,*
> > >
> > >
> > > $$ \mathbb{E}[F(\bar{z}^T)] - F(z*) \leq \mathcal{O}(\sqrt{\frac{((\frac{4N}{(N-1)|S_t|}(1-\frac{|S_{t}|}{N})+1)G^2 +\sigma^2)\mathbb{E}||z^{0}-z*||^2}{T}} + \sqrt[3]{\frac{L \mathbb{E}||z^{0}-z*||^2}{ T^2}[(\frac{(1-\lambda)^2}{3}+\frac{\lambda^4}{(1-\lambda)^2}(\frac{4N}{(N-1)|S_t|}(1-\frac{|S_{t}|}{N})+1))G^2 +(\frac{(1-\lambda)^2}{3}+\frac{\lambda^4}{(1-\lambda)^2})\sigma^2]} + \frac{L}{T} \mathbb{E}||z^{0}-z*||^2)$$
> > >
> > >
> > > We proved them using the lemmas used for non-convex case (Appendix A.1 in the revision), and lemma 1 and lemma 2 in SCAFFOLD [Karimireddy et al., 2020]. We will include the details of the proofs in the final version.
> > >
> > > In our theoretical analysis of FedACG in strongly convex and convex functions, as with non-convex cases, we also proved that FedACG converges under $0 \leq \lambda < 1$. Please refer to our response **A1.2** to reviewer KHhq for more discussions on the convergence of FedACG.

---

> > > > ### Author Response · Authors · 2022-12-09
> > > > **Look forward to hearing your feedback**
> > > >
> > > > We appreciate your time and efforts in reviewing our paper, and we have revised the manuscripts and provided detailed responses to address your concerns. As the discussion period is only a few days left, we hope you respond to our rebuttal if you have any remaining concerns or questions, or if all your concerns have been resolved well. We would be happy to continue the discussion during the remaining discussion period.

---

### Official Review · Reviewer_KHhq · 2022-10-25

**Confidence:** 4
**Correctness:** 3
**Technical Novelty And Significance:** 2
**Empirical Novelty And Significance:** 3
**Recommendation:** 5

**Clarity, Quality, Novelty And Reproducibility:**

The paper proposes novel federated learning framework, and the supplementary contains demo code and the detailed hyperparameter setting, thus is good at reproducibility.

Several comments about the clarity and quality:
1. In Table 2, in the list of the accuracy of CIFAR-10, MOON has a 64.55% accuracy while FedACG has 63.70%, thus the best accuracy is obtained by MOON.
2. In what form of accelerated client gradient is in Table 4? It is not very clear and it is better to have more detailed illustrations
3. Some typos: In the last paragraph of the related work, “FedCAM”—>”FedCAMS”.
After Equation 4, there may be one redundant letter “s” after $\{l_i(\cdot)\}_{I=1}^N$.


**Strength And Weaknesses:**

Strengths:
1. The proposed method seems effective. The introduced momentum term of the global client makes the local model close to the global one. This method does not involve additional communication costs thus it is communication efficient.
2. The paper shows a complete experimental setup covering several cases that effectively supports the proposed FedACG.

Weaknesses:
1. From the Algorithm and Theorem 1, it seems the client computes full gradients. Hence the convergence analysis lacks consideration under stochastic gradient variance.
2. There lacks of nonconvex convergence analysis which is more practical in the real world.



**Summary Of The Paper:**

This paper solves the bottleneck of slow and unstable convergence issues in heterogeneous federated learning especially when the client participation ratio is low. The paper proposes a novel framework that can improve consistency to alleviate the heterogeneous issue and then accelerate the convergence of federated learning.

**Summary Of The Review:**

1 . The theoretical result seems a bit weak. It only contains a convex convergence result in the full batch setting which is not practical.  Also it is not clear whether the obtained convergence rate is better than other methods.
2. While the main story is about improving FL when the participation ratio is low, in the theoretical result, I didn’t find the term corresponding to the ratio. Did the author only consider the full participation setting in the theorem?
3. Why do we have the constraint that $0.5 < \lambda < 1$? Any intuitions?

This paper provides novel methods for improving slow and unstable convergence in federated learning, and solid experimental results back up the method. The convergence analysis can be further improved with detailed discussions.

---

> ### Author Response · Authors · 2022-11-09
> **Response to Reviewer KHhq**
>
> Thank you for your meaningful feedback. We will improve the completeness of the work by accepting your suggestion. We address your comments here.
>
> **Q1.1\. Minor concerns for clarification**
>
> Q1.1.1\:  The meaning of the term “accelerated client gradient” in Table 4.
>
> A1.1.1\: The term “accelerated client gradient” denotes that each participating client runs its local optimization using the momentum-integrated model $\theta^{t-1} + \lambda m^{t-1}$ not the current global model $\theta^{t-1}$.
>
> Q1.1.2\: Typos
>
> A1.1.1\:  Thanks for carefully reading our paper and helping us identify these typos. We have fixed them in the revision.
>
> **Q1.2\. Convergence analysis**
>
> A1.2\: We will provide a detailed analysis and corresponding discussion for the convergence of FedACG, including the analysis of FedACG with non-convex functions.

---

> > ### Author Response · Authors · 2022-11-19
> > **Further Responses to Reviewer KHhq**
> >
> > **Q1.2. Convergence analysis**
> >
> > A1.2: As promised in our previous posting, we have proved the convergence of FedACG for non-convex functions, taking into account the stochastic variance, and updated the details of the proof in our revision (Appendix A).
> >
> > Here we present the theoretical convergence rate of FedACG for non-convex functions.
> >
> > **Theorem 1.** *(Convergence of FedACG) Suppose that local functions $f_{i}$ are non-convex and $L$-smooth.
> > Let $z^t = \theta^t + \frac{\lambda}{1 - \lambda} m^t$ for any $0 \leq \lambda < 1$.
> > Then, by setting $\eta = \text{min} (\frac{1-\lambda}{2 LK}, \frac{C}{\sqrt{T+1}})$, FedACG satisfies,*
> >
> >  $$ \text{min}_{t=0, \ldots, T}~\mathbb{E} \| \| \nabla F(\theta^t + \lambda m^t)\| \|^2 \leq  \frac{2(F(z_0) )-F*)(1-\lambda)}{T+1} \text{max}(\frac{2LK}{1-\lambda},\frac{\sqrt{T+1}}{C})+\frac{C}{\sqrt{T+1}} B'',$$
> >
> > where
> > $$ B'' =\frac{1}{(1-\lambda)K}(
> >  ( (1+\frac{L^2K^4}{3})(1-\lambda) + \frac{ \lambda^4 L K^2}{2(1-\lambda)^2} + (1+ \frac{4N}{|S_t|(N-1)}(1-\frac{|S_t|}{N}))(LK^2 + \frac{L\lambda^4 K^2}{2(1-\lambda)^2})
> > )G^2 +  \frac{LK^2}{2} \big(2+\frac{\lambda^4}{(1-\lambda^2)} \big)\sigma^2.
> > )$$
> >
> > Based on this result we address your comments as follows.
> >
> > **Consideration of stochastic variance**
> >
> > Our new proof for the non-convex setting considers the stochastic variance (Lemma 1 in Appendix), which means that the convergence rate of our algorithm is at least $\mathcal{O}(\frac{1}{\sqrt{T}})$ with the stochastic variance.  We believe we can even derive convergence of FedACG with convex functions using the similar technique we used for the proof with non-convex functions.
> >
> > **Comparison with other methods**
> >
> > The convergence rate of our algorithm on the non-convex setting is $\mathcal{O}(\frac{1}{\sqrt{T}})$, which matches the rate of FedAvg, FedADAM, and SCAFFOLD. Note that the convergence rate of FedAvg on the non-convex setting is offered by [Karimireddy et al., 2020]. Although FedDyn and FedDC are reported to have a better convergence rate in their proof, they do not consider the stochastic variance. Most of all, FedACG outperforms all the compared methods including FedDyn and FedDC in our experiments.
> >
> > **Constraints of momentum weight $\lambda$**
> >
> > As presented in Theorem 1 of Appendix in the revised manuscript, we show the convergence rate of FedACG under the constraint $0 \leq \lambda < 1$ in the non-convex case.
> >
> > **Reference**
> >
> > [Karimireddy et al., 2020] Sai Praneeth Karimireddy et al. Scaffold: Stochastic controlled averaging for on-device federated learning. In ICML, 2020.

---

> > > ### Author Response · Authors · 2022-11-24
> > > **Convergence of FedACG for Strongly Convex and General Convex Functions**
> > >
> > > **Q1.2. Convergence analysis**
> > >
> > > A1.2. As promised in our previous posting, we have proved the convergence of FedACG for convex and strongly convex functions, taking into account the stochastic variance,
> > >
> > > **Strongly convex:**
> > >
> > > **Theorem A.** *Suppose that local functions $F_{i}$ are strongly convex ($\mu > 0$) and $L$-smooth. Let $z^t = \theta^t + \frac{\lambda}{1 - \lambda} m^t$. For $\eta \leq  \frac{1-\lambda}{2LK}$ and $ 0 \leq \lambda <1$,*
> > >
> > > $$ \mathbb{E}[F(\bar{z}^T)] - F(z*) \leq \tilde{\mathcal{O}}(\frac  { (\frac{4N}{(N-1)|S_t|}(1-\frac{|S_{t}|}{N})+1)G^2 + \sigma^2}
> > > {\mu T} + \frac{L}{\mu^2 T^2}[(\frac{(1-\lambda)^2}{3}+\frac{\lambda^4}{(1-\lambda)^2}(\frac{4N}{(N-1)|S_t|}(1-\frac{|S_{t}|}{N})+1))G^2 +(\frac{(1-\lambda)^2}{3}+\frac{\lambda^4}{(1-\lambda)^2})\sigma^2]+\mu \mathbb{E}||z^{0}-z*||^2 exp(- \frac{\mu}{L}T) ) $$
> > >
> > >
> > > **General convex:**
> > >
> > > **Theorem B.** *Suppose that local functions $F_{i}$ are general convex ($\mu = 0$) and $L$-smooth. Let $z^t = \theta^t + \frac{\lambda}{1 - \lambda} m^t$. For $\eta \leq  \frac{1-\lambda}{2LK}$ and $ 0 \leq \lambda <1$,*
> > >
> > >
> > > $$ \mathbb{E}[F(\bar{z}^T)] - F(z*) \leq \mathcal{O}(\sqrt{\frac{((\frac{4N}{(N-1)|S_t|}(1-\frac{|S_{t}|}{N})+1)G^2 +\sigma^2)\mathbb{E}||z^{0}-z*||^2}{T}} + \sqrt[3]{\frac{L \mathbb{E}||z^{0}-z*||^2}{ T^2}[(\frac{(1-\lambda)^2}{3}+\frac{\lambda^4}{(1-\lambda)^2}(\frac{4N}{(N-1)|S_t|}(1-\frac{|S_{t}|}{N})+1))G^2 +(\frac{(1-\lambda)^2}{3}+\frac{\lambda^4}{(1-\lambda)^2})\sigma^2]} + \frac{L}{T} \mathbb{E}||z^{0}-z*||^2)$$
> > >
> > >
> > > We proved them using the lemmas used for non-convex case (Appendix A.1 in the revision), and lemma 1 and lemma 2 in SCAFFOLD [Karimireddy et al., 2020]. We will include the details of the proofs in the final version.
> > >
> > > Based on these results we address your comments as follows.
> > >
> > > **Consideration of stochastic variance**
> > >
> > > Our new proofs considering the stochastic variance show that the asymptotic convergence rates of FedACG are $\tilde{\mathcal{O}}(\frac{1}{T})$ and $\mathcal{O}(\frac{1}{\sqrt{T}})$ for the strongly convex and general convex functions, respectively.
> > >
> > > **Comparison with other methods**
> > >
> > > The asymptotic convergence rates of our algorithm on the strongly convex and convex functions match the rates of SCAFFOLD. As with the non-convex case, although FedDyn and FedDC are reported to have better convergence rates in their proof, they do not consider the stochastic variance.
> > >
> > >
> > > **discussion about the participation rate.**
> > >
> > > As illustrated in Theorem 1, A, and B, our new proofs for the three cases (strongly convex, convex, and non-convex) considering the stochastic variance show faster convergence as the participation rate increases.
> > >
> > > While it is difficult to provide theoretical evidence that our algorithm is more effective at low participation rates than other algorithms, as mentioned in the main paper, the proposed algorithm has the practical advantage of not using local states for training. For example, FedDyn, FedDC, and MOON should store and utilize the local states that can be easily outdated due to low participation rates, which can eventually hinder model training. In contrast, since FedACG does not store local state, it is better suited in situations where clients rarely rejoin training or where participating clients change dynamically. Results in Table 2 and Table 3 in the main paper demonstrate the effectiveness of FedACG in these realistic federated settings.
> > >
> > >
> > > **Constraints of momentum weight**
> > >
> > > Similar to the results for non-convex functions, Theorem A and B show that FedACG converges under the constraint $0 \leq \lambda < 1$ in the convex and strongly convex functions, respectively. Note that $\lambda$ is the coefficient by which the global momentum is updated exponentially, and the corresponding constraint indicates that FedACG converges under the same conditions as those used in general SGD with momentum.

---

> > > > ### Comment · Reviewer_KHhq · 2022-11-30
> > > > **Further Question**
> > > >
> > > > I thank the authors for their responses. For the nonconvex convergence results provided here, I notice that it is with respect to $\theta_t + \lambda m_t$. However, when I look back at the algorithm again, I feel that your FedACG algorithm is essentially FedAvg with a larger (larger than 1) global step size (if we rewrite the algorithm and put the momentum step back at the end of epoch rather than the start the of each epoch. If your result is with respect to $\theta_t + \lambda m_t$, it seems to be the standard FedAvg convergence result under a specially chosen learning rate.
> > > >
> > > > If so, can the authors further explain why the proposed algorithm is able to achieve better performances (rather than using a different hyperparameter)?

---

> > > > > ### Author Response · Authors · 2022-11-30
> > > > > **Similarity with FedAvg**
> > > > >
> > > > > Thank you for your response.
> > > > >
> > > > > The server update in FedACG is different from that of FedAvg with the increased (larger than 1) global learning rate because, in FedACG, the directions of the global momentum $m^t$ and local updates $\Delta^t$ are different. Unlike fedAvg, which starts local updates from the global model $\theta^{t-1}$, the clients in FedACG start their local updates at the auxiliary model $\theta^{t-1}+\lambda m^{t-1}$, so, as illustrated in Figure 1 and Eq. (3) in the main paper, the global momentum $m^t$ is defined recursively as the sum of the previous momentum $m^{t-1}$ and the current local updates $\Delta^t$, *i.e*., $m^t = \theta^t - \theta^{t-1} = \Delta^t + \lambda m^{t-1}$.
> > > > >
> > > > > On the other hand, we proved the convergence of FedACG with respect to the auxiliary point $\theta^t + \lambda m^t$ for the non-convex case for two reasons. First, our proof considers that each participating client actually computes its local gradient at the accelerated point $\theta^t + \lambda m^t$, not at the global model $\theta^t$. Second, the convergence with respect to the auxiliary point $\theta^t + \lambda m^t$ naturally guarantees the convergence with respect to the global model $\theta^t$ since the initial point and the convergence point of this auxiliary point are the same as the global model: $m^0 = 0$ and $m^{\infty} \rightarrow 0$.

---

> > > > > > ### Comment · Reviewer_KHhq · 2022-11-30
> > > > > > **Similarity with FedAvgM?**
> > > > > >
> > > > > > Thanks for the response. Based on the author's response, it seems to be a large global learning rate version of FedAvg with momentum. Again, if so, why the proposed algorithm is able to achieve better performances compared to FedAvgM?
> > > > > >
> > > > > > For convergence, since your output is $\theta_t$, your convergence result should also focused on $\theta_t$? I wonder if the authors change the algorithm output to be $\theta_t + \lambda m_t$, whether the performance is the same.

---

> > > > > > > ### Author Response · Authors · 2022-12-01
> > > > > > > **Further response to reviewer KHhq**
> > > > > > >
> > > > > > > Thank your for your response. We answer to your questions as below.
> > > > > > >
> > > > > > > **Q. Similarity with FedAvgM with a large global learning rate**
> > > > > > >
> > > > > > > A: FedACG is different from FedAvgM with a large global learning rate for two reasons. First,
> > > > > > > the server in FedACG just averages the local models to get the global model for the next round, so FedACG does not have any term or hyperparameter for controlling the global learning rate. Second, the main difference between FedACG and FedAvgM lies in the initial point where each client starts its local updates. To clarify the difference between FedACG and FedAvgM, we provide detailed procedures for each algorithm below. Unlike FedAvgM, where the server broadcasts the previous global model $\theta^{t-1}$ to the clients, in FedACG, the server broadcasts the interim model $\theta^{t-1} + \lambda m^{t-1}$ to the clients.
> > > > > > > Note that, since the interim model $\theta^{t-1} + \lambda m^{t-1}$ is not the output global model, we cannot simply put this step back at the end of the epoch.
> > > > > > > In FedACG, each client can lookahead the impact of the global momentum on its local loss function and adaptively compute the local gradient.
> > > > > > > Please refer to response [**A3.1** to reviewer Ym7U](https://openreview.net/forum?id=de-_FHXQ4--&noteId=f-XkbGbRxYu) and [**A4.1** to reviewer fVhK](https://openreview.net/forum?id=de-_FHXQ4--&noteId=N9qONAkbC4f) for detailed discussions and experiments about the effectiveness of this strategy.
> > > > > > >
> > > > > > >
> > > > > > > ***Procedure of FedACG (Algorithm 2 and Figure 7 in the revision)***
> > > > > > >
> > > > > > > 1. The server broadcasts **an interim parameter $\theta^{t-1} + \lambda m^{t-1}$** to the clients. *i.e.,* $\theta^t_{i,0} = \theta^{t-1} + \lambda m^{t-1}$.
> > > > > > >
> > > > > > > 2. Each client takes its local optimization step **at the interim parameter $\theta^{t-1} + \lambda m^{t-1}$**, and sends resultant local updates $\Delta^t_i = \theta^t_{i,K} - \theta^t_{i,0}$ to the server.
> > > > > > >
> > > > > > > 3. The server averages the local updates $\Delta^t = \sum_{i \in \mathcal{S}_t} \omega_i \Delta^t_i$ and calculates the global momentum $m^t = \lambda m^{t-1} + \Delta^t$.
> > > > > > >
> > > > > > > 4. The server updates the output global model $\theta^t$ using the global momentum, *i.e.,* $\theta^{t} = \theta^{t-1} + m^t$.
> > > > > > >
> > > > > > > ***Procedure of FedAvgM (Algorithm 3 and Figure 8 in the revision)***
> > > > > > >
> > > > > > > 1. The server broadcasts **the previous global model $\theta^{t-1}$** to the clients. *i.e.,* $\theta^t_{i,0} = \theta^{t-1}$.
> > > > > > >
> > > > > > > 2. Each client takes its local optimization step **at the previous global
> > > > > > > model $\theta^{t-1}$**, and sends resultant local updates ${\Delta}^t_i = \theta^t_{i,K} - \theta^t_{i,0}$ to the server.
> > > > > > >
> > > > > > > 3. The server averages the local updates ${\Delta}^t = \sum_{i \in \mathcal{S}_t} \omega_i {\Delta}^t_i$ and calculates global momentum ${m}^t = \lambda {m}^{t-1} + {\Delta}^t$.
> > > > > > >
> > > > > > > 4. The server updates the output global model $\theta^t$ using the global momentum, *i.e.,* $\theta^{t} = \theta^{t-1} + {m}^t$.
> > > > > > >
> > > > > > >
> > > > > > > **Q. Shouldn't the convergence result of FedACG be with respect to $\theta$?**
> > > > > > >
> > > > > > > A: As mentioned in the previous response, the convergence of FedACG with respect to the $\theta^t$ can be naturally guaranteed by the convergence of FedACG with respect to the auxiliary point $\theta_t + \lambda m_t$ since the auxiliary point has the identical initial point and convergence point to the output global model $\theta^t$. This technique is widely used to prove the convergence of various optimization algorithms [Yan et al., 2018, Acar et al., 2021]. For example, FedDyn [Acar et al., 2021] have shown the convergence of the proposed algorithm with respect to $\gamma^t$ while the output global model is $\gamma^t - \frac{1}{\alpha}h^t$.
> > > > > > >
> > > > > > > **Q. Results of FedACG with output $\theta_t + \lambda m_t$**
> > > > > > >
> > > > > > > A: We observe empirically that even if we change the output global model as $\theta^t + \lambda m^t$, FedACG shows consistent accuracy at 1K rounds (62.02% for $\theta_t + \lambda m_t$ and 62.51% for $\theta_t$) in the moderate-scale experiment on CIFAR100 with Dirichlet parameter 0.3 (Table 1(a) in the main paper).
> > > > > > >
> > > > > > > **Reference**
> > > > > > >
> > > > > > > [Yan et al., 2018]  Yan Yan et al., A Unified Analysis of Stochastic Momentum Methods for Deep Learning. In IJCAI, 2018.

---

> > > > > > > > ### Comment · Reviewer_KHhq · 2022-12-04
> > > > > > > > **Thanks for the clarification but still some concerns**
> > > > > > > >
> > > > > > > > I thank the authors for the clarification. Yet my main concern is still this relationship with FedAvgM and how the performance gain is obtained.
> > > > > > > >
> > > > > > > > First, $\lambda$ here is roughly the "global learning rate" so it does not really get rid of the global lr. Second, I understand and agree that the current algorithm is not equivalent to FedAvgM, but it is because the output global model is different (and thus we cannot simply put the first step back at the end of the epoch). However, if the output global model is $\theta_t + \lambda m_t$, we can actually put it back and make it a special case of FedAvgM.
> > > > > > > >
> > > > > > > > At this point, I am assuming that if the performance gain over FedAvgM is not from tuning the global learning rate, it must come from this choice of output global model. So my previous question is actually asking, why such a change in output model will give you better results. Yet the authors told me that empirically changing this output model does not have much difference, which makes me confused on what is the actual key ingredient in the formulation.

---

> > > > > > > > > ### Author Response · Authors · 2022-12-04
> > > > > > > > > **The main difference is the initial point where each client starts to learn**
> > > > > > > > >
> > > > > > > > > Thank you for your active feedback.
> > > > > > > > >
> > > > > > > > > We believe that there are still misunderstandings. Please focus on steps 1 and 2 of both FedACG and FedAvgM, which are related to the initializations of client models. In FedACG, each client starts to learn from $\theta^{t-1} + \lambda m^{t-1}$ while the clients with FedAvgM start at $\theta^{t-1}$. By making the starting point of learning different from FedAvgM, we obtain a more robust model and achieve better results. Note that steps 3 and 4 of both algorithms are identical. Also, our output global model is not $\theta^{t} + \lambda m^{t}$ but $\theta^t$ as described in the algorithm pseudocode. This is the same in FedAvgM.
> > > > > > > > >
> > > > > > > > > We made minor revisions in the algorithm description part of [our previous answer](https://openreview.net/forum?id=de-_FHXQ4--&noteId=Pr4uTyGK-1Y). Please let us know if you still have questions or concerns.
> > > > > > > > >
> > > > > > > > > We truly appreciate your feedback and hope this answer figures out the issues.

---

> > > > > > > > > > ### Comment · Reviewer_KHhq · 2022-12-05
> > > > > > > > > > **Thanks for the clarificiation**
> > > > > > > > > >
> > > > > > > > > > I agree we can also view it as changing the starting point compared to FedAvgM.  Thanks for the clarification.  Then, following your story, I wonder why the change of starting point leads to empirical better performances. Simply saying "by making the starting point of learning different" is not a convincing enough statement.
> > > > > > > > > >
> > > > > > > > > > Second, my previous posts are viewing it from a different path by changing FedAvgM's output point. Specifically, from FedAvgM algorithm, if you set $\theta_{t} = \theta_{t-1} + (1+\lambda) m_t$ and the output as $\theta_{t-1} +  m_t$ rather than $\theta_t$, I believe you also get the same algorithm as FedACG. In this view, from FedACG, if we add extra $\lambda m_t$ on the output, it will come back to FedAvgM. Hope this makes my previous question clear.

---

> > > > > > > > > > > ### Author Response · Authors · 2022-12-05
> > > > > > > > > > > **Further responses to Reviewer KHhq**
> > > > > > > > > > >
> > > > > > > > > > > Thank you for clarification about your previous question.
> > > > > > > > > > >
> > > > > > > > > > > **FedACG is a different algorithm than the modified version of FedAvgM mentioned in the previous comment**
> > > > > > > > > > >
> > > > > > > > > > > We believe that there are still misunderstandings. We cannot view changing the starting point in FedACG as changing FedAvgM’s output point (setting $\theta^t = \theta^{t-1} + (1+\lambda)m^t$ and setting the output $\theta^{t-1} + m^t$), since such a modification results in the totally different algorithm than FedACG. Specifically, if you resolve the recursion of the initial point that clients start learning in round t, clients with FedACG start at $\theta^0 + \sum_{i=1}^{t} m^{i} + \lambda m^t$, while the clients with the modified FedAvgM start at $\theta^0 + (1+\lambda) \sum_{i=1}^{t} m^{i}$. This difference results in different local gradients $\Delta^{t+1}$ and global momentum $m^{t+1}$, which indicates that these two algorithms are different.
> > > > > > > > > > >
> > > > > > > > > > > **Why FedACG outperforms FedAvgM**
> > > > > > > > > > >
> > > > > > > > > > > It is generally good to use global momentum, which is the exponential sum of previous global gradients, for the server update. However, the stability of the heterogeneous federated learning algorithm depends on where and how momentum is used, and our learning strategy is simple yet more effective to handle this issue than FedAvgM. Since only a subset of the clients participates in the training for each round and the norm of local updates may be large due to K local gradient steps, high data heterogeneity in the federated learning can cause the global momentum to be noisy. Unlike FedAvgM, where the only server knows the global momentum, in FedACG, the clients, the actual learners, start their local training at the anticipated point $\theta^{t-1}+ \lambda m^{t-1}$ and can make up for the previous global momentum $\lambda m^{t-1}$ through their local training. This difference in the initial point seems to allow FedACG to change $m^t$ more responsive way, letting it behave more stable than FedAvgM in many federated learning settings. The effect of the difference for each communication round may be accumulated as the communication progresses.

---

> > > > > > > > > > > > ### Comment · Reviewer_KHhq · 2022-12-06
> > > > > > > > > > > > **Thanks**
> > > > > > > > > > > >
> > > > > > > > > > > > "FedACG is a different algorithm than the modified version of FedAvgM mentioned in the previous comment"
> > > > > > > > > > > >
> > > > > > > > > > > > ==> Thanks for the clarification. Now I see that is indeed different
> > > > > > > > > > > >
> > > > > > > > > > > > "Why FedACG outperforms FedAvgM"
> > > > > > > > > > > >
> > > > > > > > > > > > ==> Your argument is that changing this starting point would make it behave more stable. Can you define this "stableness" in terms of concrete formulations and verify this stableness empirically? Currently, the only main drawback of this paper is that it is not very clear on how the advantage is obtained.

---

> > > > > > > > > > > > > ### Author Response · Authors · 2022-12-09
> > > > > > > > > > > > > **Why FedACG outperforms FedAvgM**
> > > > > > > > > > > > >
> > > > > > > > > > > > > Thank you for your feedback.
> > > > > > > > > > > > >
> > > > > > > > > > > > >
> > > > > > > > > > > > > To further analyze the effectiveness of FedACG over FedAvgM in the heterogeneous federated learning, we plot the accuracy of FedACG and FedAvgM on CIFAR-10 in the moderate-scale setting without smoothing (exponential moving average), and we provide the results in this [link](https://anonymous.4open.science/r/paper1852_clarification-B831/cifar10_moderate.png). For the experiments, we set the momentum coefficient as 0.85 for both algorithms. We first observe that FedACG consistently outperforms FedAvgM. Furthermore, If we calculate the averaged variance between the current accuracy $\text{Acc}^t$ and the simple moving average $\text{Acc}^t_{\text{SMA}}$ (we set $n$ to 51) over 1000 rounds of communication, i.e., $\frac{1}{T} \sum_{t=0}^{T-1} (\text{Acc}^t-\text{Acc}^t_{\text{SMA}})^2$, the averaged variance of FedACG is 2.26, while that of FedAvgM is 10.30. The results verify that FedACG converges with a smaller level of oscillation than FedAvgM throughout the training procedure. We believe that this is because FedACG allows each client’s update to be responsive to the previous momentum by making the initial point of the local update start at the anticipated point $\theta^{t-1} + \lambda m^{t-1}$.
> > > > > > > > > > > > >
> > > > > > > > > > > > >
> > > > > > > > > > > > > We sincerely appreciate your efforts in reviewing our paper. We have responded to your comments and faithfully reflected them in the revision. We hope that our revision and responses have addressed all your concerns, and hope you reflect this in your final review and the score. If you have any remaining concerns then feel free to let us know. We would be happy to continue the discussion during the remaining discussion period.

---

### Author Response · Authors · 2022-11-24
**General Response and Revision Summary**

We thank all reviewers for your constructive comments and suggestions, which are greatly helpful for improving the quality of our paper. We have responded to the concerns regarding the novelty of the proposed method, theoretical analysis, and experimental results. If you have any remaining concerns then feel free to let us know. We would be happy to continue the discussion if there are additional questions.

We first emphasize the contributions and strengths of our paper generally agreed by reviewers:

* We propose a communication-efficient federated learning framework that addresses the slow and unstable convergence issues due to client heterogeneity in federated learning.

* Our algorithm is better suited for realistic settings of federated learning; it does not require additional communication cost per round, and it does not require additional memory overhead for the clients.

* Extensive experiments on more realistic benchmarks and challenging federated settings demonstrate the effectiveness of the proposed method in terms of communication efficiency and robustness to client heterogeneity.

We summarize the key points from our responses:

* We have clarified the fundamental differences between our proposed algorithm and other federated learning algorithms (FedProx and FedAvgM): [Response A2.1 to reviewer bEuE](https://openreview.net/forum?id=de-_FHXQ4--&noteId=9uQ_u4oknL) for FedProx, and [Response to reviewer KHhq](https://openreview.net/forum?id=de-_FHXQ4--&noteId=Pr4uTyGK-1Y), [Response A3.1 to reviewer Ym7U](https://openreview.net/forum?id=de-_FHXQ4--&noteId=f-XkbGbRxYu), and [Response A4.1 to reviewer fVhK](https://openreview.net/forum?id=de-_FHXQ4--&noteId=N9qONAkbC4f) for FedAvgM.

    (Updated) We provided further discussions on why the different initial point in FedACG is effective in heterogeneous federated learning. [Response to reviewer KHhq](https://openreview.net/forum?id=de-_FHXQ4--&noteId=GwlfIvqJH9q) We also reported empirical analysis that supports our claim. [Further response to reviewer KHhq](https://openreview.net/forum?id=de-_FHXQ4--&noteId=puFMstGQswZ)


* We have proved the convergence of the proposed optimization technique for strongly convex, convex, and non-convex functions in a more practical setting considering stochastic variance, and also provided detailed discussions based on the theorems. [Response A1.2 to reviewer KHhq](https://openreview.net/forum?id=de-_FHXQ4--&noteId=VzPtjprttq), [Response A2.4 to reviewer bEuE](https://openreview.net/forum?id=de-_FHXQ4--&noteId=xWiAElqVdV), and [Response A4.3 to reviewer fVhK](https://openreview.net/forum?id=de-_FHXQ4--&noteId=glwmxFPmBN)

* We evaluated our algorithm on another task (language modeling on ShakeSpeare), using a different backbone (LSTM) in the presence of high data imbalance. This strengthens the generalization ability of our framework on different tasks, model architectures, and data heterogeneity. [Response A3.3 and A3.4 to reviewer Ym7U](https://openreview.net/forum?id=de-_FHXQ4--&noteId=hUA_xBof0C)

---

### Decision · Program_Chairs · 2023-01-20

**Decision:**

Reject

**Justification For Why Not Higher Score:**

Insufficient evidence to support all of the claims, especially given that the proposed method is very similar to prior work in the literature.

**Justification For Why Not Lower Score:**

N/A

**Metareview: Summary, Strengths And Weaknesses:**

This paper propose an approach to accelerate federated learning (FL) called FedACG. The approach is closely related to some well-known previous work (most notably FedProx and FedAvgM). The paper provides a theoretical convergence guarantee in the case of smooth convex losses, and it also provides experiments illustrating the promise of the proposed approach on the MNIST, CIFAR-10, and Tiny-ImageNet datasets.

The experimental results are promising.

At the same time, this is an area that has received a lot of attention over the past few years, and the proposed approach is very similar to others in the literature. The reviewers had a difficult time understanding the key differences from previous work that allow this approach to perform so much better. After much discussion among the reviewers and authors, this was somewhat clarified. However, the reviewers and AC were still not convinced that we understood why the subtle differences result in such marked performance improvements. This was the main weakness leading towards the recommendation that this paper be rejected. There was also an extensive discussion about these points involving the AC and all of the reviewers.

There are two key ways in which the paper could be strengthened to have a better chance of being accepted next time:
1. The writing could be improved to better put in context the contributions of this work in relation to previous work, especially clarifying the differences from previous methods like FedAvgM.
2. Additional experiments or analysis should be added to convince the reader why the specific changes introduced in this paper (i.e., differences from FedAvgM, FedProx, and other nearby methods) lead to the substantial improvements observed in the experiments.

**Summary Of Ac-Reviewer Meeting:**

We had a virtual meeting to discuss this paper. All reviewers attended and contributed to the discussion.

We discussed the merits and weaknesses of the paper, most of which were already summarized above.

Main Merits: Promising experimental results

Main Weaknesses:
* Lots of confusion about the relationship to previous work based on the initial submission, leading to the perception/conclusion that the contribution is incremental.
* Lack of evidence (either in experiments or theory) to support why the proposed approach (which involves small, subtle changes to prior work) results in seemingly substantial performance improvements.

When weighing these points to reach a decision, the weaknesses clearly outweighed the merits in this case, resulting in the recommendation to reject the paper.

After the discussion, the reviewers also all agreed that rejecting this paper was the right decision given its current state.